# Reward probability and timing uncertainty alter the effect of dorsal raphe serotonin neurons on patience

Katsuhiko Miyazaki [1], Kayoko W. Miyazaki[1], Akihiro Yamanaka[2], Tomoki Tokuda[3], Kenji F. Tanaka[4] & Kenji Doya[1]

Recent experiments have shown that optogenetic activation of serotonin neurons in the dorsal raphe nucleus (DRN) in mice enhances patience in waiting for future rewards. Here, we show that serotonin effect in promoting waiting is maximized by both high probability and high timing uncertainty of reward. Optogenetic activation of serotonergic neurons prolongs waiting time in no-reward trials in a task with 75% food reward probability, but not with 50 or 25% reward probabilities. Serotonin effect in promoting waiting increases when the timing of reward presentation becomes unpredictable. To coherently explain the experimental data, we propose a Bayesian decision model of waiting that assumes that serotonin neuron activation increases the prior probability or subjective confidence of reward delivery. The present data and modeling point to the possibility of a generalized role of serotonin in resolving trade-offs, not only between immediate and delayed rewards, but also between sensory evidence and subjective confidence.

[1] Neural Computation Unit, Okinawa Institute of Science and Technology Graduate University, Okinawa 904-0495, Japan. [2] Department of Neuroscience II, Research Institute of Environmental Medicine, Nagoya University, Nagoya 464-8601, Japan. [3] Mathematical and Theoretical Physics Unit, Okinawa Institute of Science and Technology Graduate University, Okinawa 904-0495, Japan. [4] Department of Neuropsychiatry, School of Medicine, Keio University, Tokyo 160-8582, Japan. These authors contributed equally: Katsuhiko Miyazaki, Kayoko W. Miyazaki. Correspondence and requests for materials should be addressed to K.M. (email: miyazaki@oist.jp)

The neuromodulator, serotonin, is extensively involved in behavioral, affective, and cognitive functions of the brain. Chemical and electrode recordings from the dorsal raphe nucleus (DRN) have shown that the activity of serotonin neurons increases when animals perform tasks requiring them to wait for delayed rewards[1–3]. Local pharmacological inhibition of DRN serotonin neural activity in rats impairs their patience in waiting for delayed rewards[4]. We recently used transgenic mice that express the channelrhodopsin-2 (ChR2) variant C128S in serotonin neurons[5,6] and showed that their selective activation in the DRN enhances the patience of mice waiting for both a conditioned reinforcer tone and a food reward[7]. A recent study also confirmed that optogenetic activation of DRN serotonin neurons enhances patience in waiting[8]. These results established a causal relationship between serotonin neural activation and patience in waiting for future rewards.

We therefore questioned whether activation of serotonin neurons always promotes waiting for delayed reward or whether its effect depends on the subject's reward prediction. In our previous optogenetic study, serotonergic activation prolonged waiting time by ~30% before the mice eventually gave up waiting[7]. Serotonin neuron activation was most effective at the time when mice decided whether to continue waiting[7]. These results suggest that cognitive status, such as the anticipation of future rewards, modulates the promotion of patience by serotonin.

In the current study, we tested whether the probability, amount, and timing uncertainty of future rewards affects promotion of patience by serotonin neuron activation. We find that serotonin effect in promoting waiting is maximized by both high-reward probability (RP) and high-reward timing uncertainty. We further propose a Bayesian decision model of waiting, which assumes serotonin neuron activation increases the prior RP to reproduce the major features of the experimental results. The model reproduces the more prominent effect of serotonin with reward timing uncertainty because the likelihood function for reward delivery has a longer tail in time. The present data and modeling suggest that serotonin neuron activation enhance patience in waiting for future rewards by increasing subjective confidence of future goals.

## Results

**Serotonin effect on waiting depends on reward probability.** Mice (seven transgenic mice and five wild-type (WT) mice) were trained to perform a sequential tone-food waiting task that required them to wait for a delayed tone (conditioned reinforcer) at a tone site and then to wait for delayed food (primary reward) at a reward site (Fig. 1a, b). In experiment 1, to examine whether the predicted probability and amount of reward affect the promotion of patience by serotonin neuron activation, we prepared six combinations of RP (75, 50, and 25%) and reward amount (1, 2, and 3 food pellets) (Supplementary Fig. 1).

In the experiment, during which 75% of the nose pokes for 3 s were rewarded with one food pellet (Supplementary Fig. 1a), waiting time in the 25% of trials with no reward (i.e., omission) was significantly longer with serotonin neuron activation (7.89 ± 0.08 s; mean ± s.e.m.) than without activation (6.95 ± 0.09 s; $t(5) = 24.05$, $P = 2.32 \times 10^{-6}$, $n = 6$ mice, paired $t$-test) (Figs. 2a and 3a; Supplementary Fig. 2). The effect was significantly seen in each of the six mice tested ($P < 0.022$, Mann–Whitney $U$-test) (Supplementary Fig. 3). We confirmed, in five WT mice, that waiting time in the blue light trials (7.36 ± 0.31 s) was not significantly different from that in the yellow light trials (7.35 ± 0.32 s; $t(4) = 0.33$, $P = 0.76$, $n = 5$ mice, paired $t$-test). In the 75% one-pellet test, we analyzed control group (WT) data with ChR2-expressing group (ChR2) in a two-way analysis of variance (ANOVA). There was a significant main effect of light (two levels within-subject factors; yellow and blue, $F(1,9) = 366.83$, $P < 10^{-6}$) but no significant main effect of group (two levels between-subject factors; ChR2 and WT, $F(1,9) = 0.062$, $P = 0.81$). There

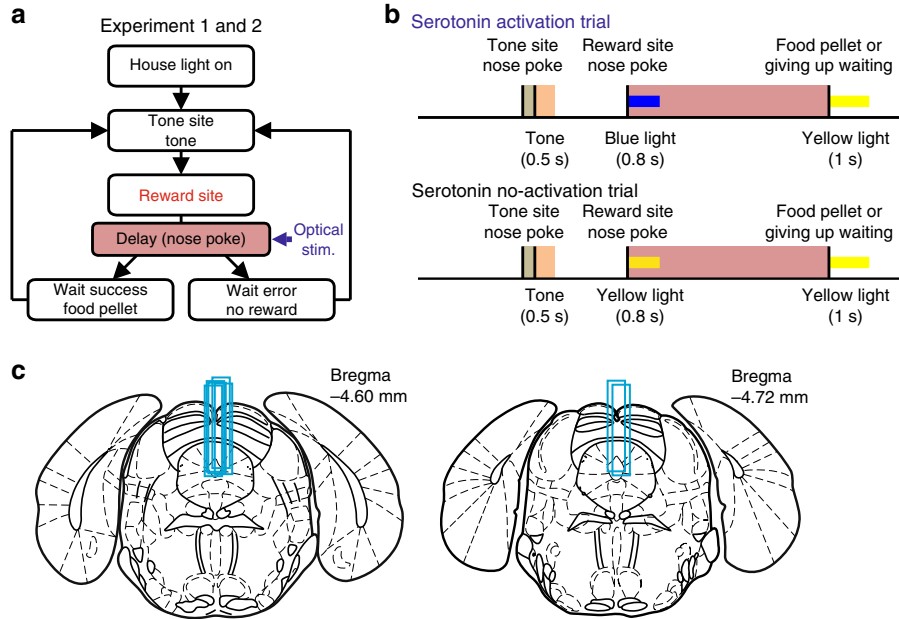

**Fig. 1** Schematic of the sequential tone-food waiting task. **a** Diagram of the test in which optogenetic stimulation was applied during the reward-delay period (experiments 1 and 2). **b** Time sequence of serotonin activation trials and serotonin no-activation trials. In serotonin activation trials, 0.8 s of blue light was delivered at the onset of the reward delay. In serotonin no-activation trials, 0.8 s of yellow light was applied at the onset of the reward delay. In each trial, 1 s of yellow light was used at the onset of food presentation or at the reward wait error. Blue and yellow bars denote blue and yellow light stimulation, respectively. Brown- and red-shaded regions denote tone- and reward-delay periods, respectively. Orange-shaded regions denote duration of tone presentation. **c** Locations of optical fibers in the DRN. Light blue bars in the DRN represent tracks of implanted optical fibers. Coronal drawings were adapted from ref. [48] with permission

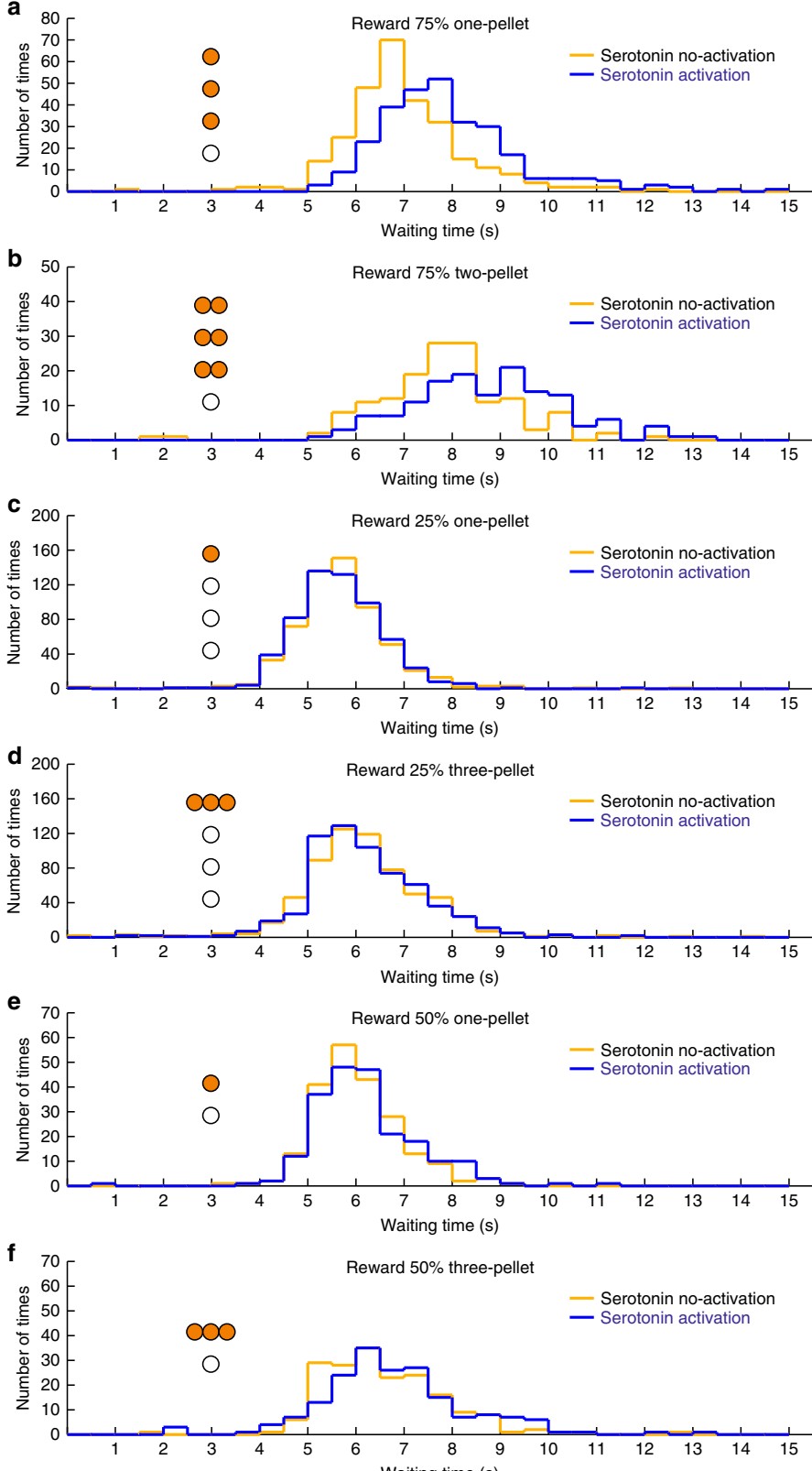

**Fig. 2** Optogenetic activation of DRN serotonin neurons enhances waiting in 75% reward tests, but not in 25 or 50% reward tests. **a** Distribution of waiting time during omission trials in the 75% one-pellet test. **b** Distribution of waiting time during omission trials in the 75% two-pellet test. **c** Distribution of waiting time during omission trials in the 25% one-pellet test. **d** Distribution of waiting time during omission trials in the 25% three-pellet test. **e** Distribution of waiting time during omission trials in the 50% one-pellet test. **f** Distribution of waiting time during omission trials in the 50% three-pellet test. Orange circles illustrate the timing and numbers of food pellets presented in rewarded trials. White circles denote omission trials

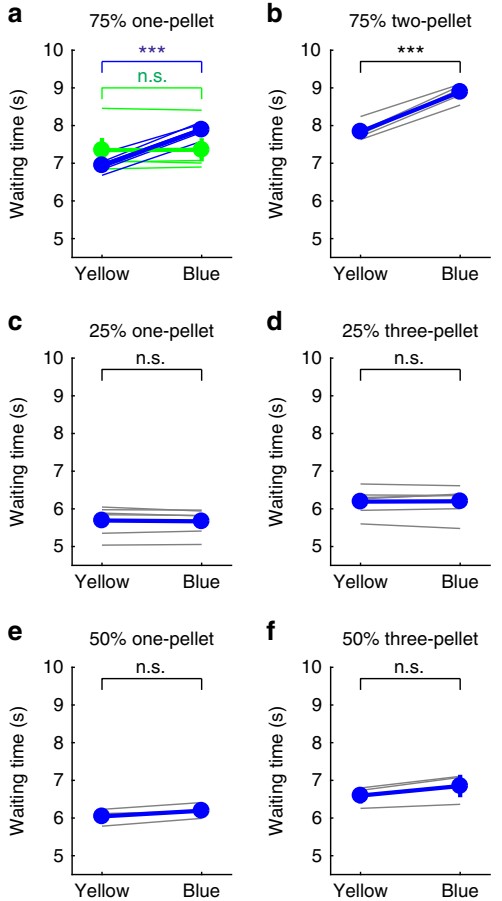

**Fig. 3** Average waiting time during omission trials in the 75, 50, and 25% reward tests. **a** Average waiting time in serotonin no-activation (yellow) and activation (blue) during the 75% one-pellet test for individual ChR2-expressing (blue thin lines) and WT (green thin lines) mice and for population of ChR2-expressing (blue line) and WT (green line) mice. **b** Average waiting time in serotonin no-activation (yellow) and activation (blue) during the 75% two-pellet test for individual ChR2-expressing mice (gray lines) and for population of mice (blue line). **c** Average waiting time in serotonin no-activation (yellow) and activation (blue) during the 25% one-pellet test for individual ChR2-expressing mice (gray lines) and for population of mice (blue line). **d** Average waiting time in serotonin no-activation (yellow) and activation (blue) during the 25% three-pellet test for individual ChR2-expressing mice (gray lines) and for population of mice (blue line). **e** Average waiting time in serotonin no-activation (yellow) and activation (blue) during the 50% one-pellet test for individual ChR2-expressing mice (gray lines) and for population of mice (blue line). **f** Average waiting time in serotonin no-activation (yellow) and activation (blue) during the 50% three-pellet test for individual ChR2-expressing mice (gray lines) and for population of mice (blue line). ***$P < 0.001$ by paired $t$-test. Error bars represent the s.e.m. In some case, the error bars are too small to be visible. n.s. not significant

was a significant main effect of interaction (light × group, $F(1,9) = 353.14$, $P < 10^{-6}$). There was a significant simple main effect of light in ChR2 ($F(1,9) = 791.90$, $P < 10^{-6}$) but no significant simple main effect of light in WT ($F(1,9) = 0.06$, $P = 0.81$) (Fig. 3a). When the reward was increased to two pellets, waiting times for omission trials became significantly longer both without serotonin neuron activation ($7.84 \pm 0.12$ s, $t(4) = 7.45$, $P = 0.0017$, $n = 5$ mice, paired $t$-test) and with ($8.89 \pm 0.11$ s, $t(4) = 5.42$, $P = 0.0056$, $n = 5$ mice, paired $t$-test) (Fig. 2a, b; Supplementary Figs. 2 and 4a, b). Again, waiting time with such

activation was significantly longer than that without ($t(4) = 14.74$, $P = 1.23 \times 10^{-4}$, $n = 5$ mice, paired $t$-test) (Fig. 3b).

In contrast, when the probability of reward delivery was reduced to 25% (Supplementary Fig. 1b), waiting time in omission trials with serotonin neuron activation ($5.67 \pm 0.16$ s) was not significantly different from that without ($5.69 \pm 0.18$ s; $t(5) = 0.89$, $P = 0.41$, $n = 6$ mice, paired $t$-test) (Figs. 2c and 3c; Supplementary Fig. 2). To examine whether the ineffectiveness of serotonin neuron activation was due to a lower expected reward value, we performed a test with a 25% reward of three pellets, in which the expected reward value was equated with that of a 75% reward of one pellet. Waiting time without serotonin neuron activation in the 25% three-pellet test ($6.20 \pm 0.18$ s) was significantly longer than that in the 25% one-pellet test ($t(5) = 11.79$, $P = 7.74 \times 10^{-5}$, $n = 6$ mice, paired $t$-test), but significantly shorter than in the 75% one-pellet test ($t(5) = 5.33$, $P = 0.0031$, $n = 6$ mice, paired $t$-test) (Fig. 2c, d; Supplementary Figs. 2 and 4c, d). However, even with a higher expected reward value, waiting time in omission trials in the 25% three-pellet test with serotonin neuron activation was not significantly different from that without serotonin neuron activation ($6.19 \pm 0.16$ s; $t(5) = 0.24$, $P = 0.82$, $n = 6$ mice, paired $t$-test) (Figs. 2d and 3d; Supplementary Fig. 2). These results show that the increased reward value in the 25% reward tests prolongs waiting time, but does not modulate the effect of serotonin in promoting waiting time.

To further examine whether the uncertainty of reward delivery affects the promotion of patience by serotonin, we introduced tests with a 50% RP, at which the uncertainty is maximized (Supplementary Fig. 1c). In both the one-pellet and three-pellet tests, serotonin neuron activation did not prolong waiting time in omission trials compared with the trials without serotonin neuron activation (one-pellet test, $6.19 \pm 0.15$ s, with activation, $6.04 \pm 0.16$ s, without activation, $t(2) = 3.36$, $P = 0.078$, $n = 3$ mice, paired $t$-test; three-pellet test, $6.85 \pm 0.30$ s, with activation, $6.60 \pm 0.21$ s, without activation, $t(2) = 3.44$, $P = 0.075$, $n = 3$ mice, paired $t$-test) (Figs. 2e, f and 3e, f; Supplementary Fig. 2). In the 50% three-pellet test, the expected reward value (1.5 pellets per trial) was equal to that in the 75% two-pellet test. Waiting time in omission trials without serotonin neuron activation in the 50% three-pellet test was significantly longer than those in the 50% one-pellet test ($t(2) = 6.89$, $P = 0.020$, $n = 3$ mice, paired $t$-test), but significantly shorter than those in the 75% two-pellet test ($t(2) = 4.86$, $P = 0.039$, $n = 3$ mice, paired $t$-test) (Supplementary Figs 2 and 4e, f). These results show that the uncertainty of reward acquisition does not facilitate waiting or the effect of serotonin neuron activation on waiting.

To quantify the effectiveness of serotonin neuron activation at promoting waiting time during omission trials, we calculated waiting time ratio (waiting time with serotonin neuron activation/ waiting time without serotonin neuron activation) for each test (Fig. 4) and performed Scheirer–Ray–Hare test with the RP and the expected reward value as explanatory variables. There was a significant main effect of the RP (three level; 75, 50, and 25%, $H(2) = 112.38$, $P < 10^{-6}$) but no significant main effect of the expected reward value (four levels; 0.25, 0.5, 0.75, and 1.5, expected pellets (EPs) per trial, $H(3) = 0.11$, $P = 0.99$).

In addition, we performed analysis based on a linear mixed model, taking mouse identity (MI) as a random effect. This approach is based on a plausible assumption that the baseline waiting time ratio may be different among mice. The result of likelihood ratio test between the model with effects of RP and EP and the model without these covariates supports the former model ($\chi^2(5) = 121.00$, $P < 10^{-6}$). Further, using the obtained model, we tested difference of mean waiting time ratios between different levels of RP and EP. Difference of means between RP 75 and 25% ($Z = 9.02$, $P < 10^{-6}$), and between 75 and 50% ($Z = 5.07$,

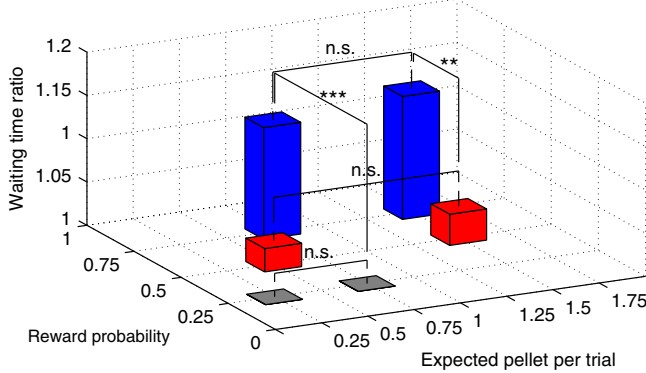

**Fig. 4** The role of serotonin in promoting patience is modulated by reward probability, but not by reward value. Waiting time ratios: 75% one-pellet test (1.13 ± 0.01, $n = 57$ tests from 6 mice), 75% two-pellet test (1.14 ± 0.02, $n = 30$ tests from 5 mice), 25% one-pellet test (1.00 ± 0.01, $n = 38$ tests from 6 mice), 25% three-pellet test (1.00 ± 0.01, $n = 41$ tests from 6 mice), 50% one-pellet test (1.03 ± 0.01, $n = 21$ tests from 3 mice), and 50% three-pellet test (1.04 ± 0.02, $n = 19$ tests from 3 mice). Waiting time ratios were significantly larger in 75% reward tests compared with tests having the same expected reward value. $**P < 0.01$, $***P < 0.001$ by post hoc Bonferroni correction. n.s. not significant. Error bars represent the s.e.m.

$P < 10^{-6}$) were significant, while the remainder of difference of means were not significant (Supplementary Table 1). Subsequently, we tested variability of waiting time ratio among mice. We compared the obtained mixed model with the model including fixed effects of RP and EP, but not a random effect of MI. To accurately evaluate likelihood ratio of two models, we generated 1000 new samples of waiting time ratios by means of a parametric bootstrap method. The variability of waiting time ratio among mice was not significant ($P = 0.553$). Lastly, we went for more detailed analysis on differences of waiting time ratios for specific combinations of RP and EP. Note that in this analysis, we did not distinguish between mice because such differences are not significant.

In each RP, reward value change did not significantly influence the waiting time ratio (75% one-pellet vs. 75% two-pellet, $P = 1.00$; 50% one-pellet vs. 50% three-pellet, $P = 1.00$; 25% one-pellet vs. 25% three-pellet, $P = 1.00$, post hoc Bonferroni correction) (Fig. 4). This result was all seen in each of the tested mice (for 75% reward, $P > 0.55$, $n = 5$ mice; for 50% reward, $P > 0.53$, $n = 3$ mice; for 25% reward, $P > 0.20$, $n = 6$ mice, Mann–Whitney $U$-test) (Supplementary Fig. 5). When we directly compared tests with different RP and same expected reward value, the waiting time ratios were significantly larger in 75% reward tests compared with same expected reward value tests (75% one-pellet vs. 25% three-pellet, $P < 10^{-6}$; 75% two-pellet vs. 50% three-pellet, $P = 0.0039$, post hoc Bonferroni correction) (Fig. 4). These results show that serotonin's effect on promoting waiting depends on the probability of delivery, but not the expected value, of future reward.

**Reward timing uncertainty alters serotonin effect on waiting**. In our previous study, the waiting time ratio was >1.3[7], whereas in experiment 1 of the current study, the waiting time ratio was ~1.1 with a 75% probability of reward. A major difference between the previous and current studies was the variability of reward delays. In our previous study, in the 75% reward trials, the reward delay was randomly set to 3, 6, or 9 s, whereas it was a constant 3 s in the current study. Thus, we hypothesized that serotonin promotes waiting more effectively when mice cannot

predict the timing of the reward delivery (timing uncertainty). In experiment 2, we prepared three reward-delay conditions with a 75% RP: (i) fixed 6 s (D6 test) (Supplementary Fig. 6a); (ii) randomly set to 4, 6, or 8 s (D4-6-8 test) (Supplementary Fig. 6b); and (iii) randomly set to 2, 6, or 10 s (D2-6-10 test) (Supplementary Fig. 6c). In all three tests, waiting time for omission trials with serotonin neuron activation was significantly longer than that without serotonin neuron activation (D6 test, 12.23 ± 0.20 s vs. 11.00 ± 0.23 s, $t(5) = 20.35$, $P = 5.30 \times 10^{-6}$, $n = 6$ mice; D4-6-8 test, 14.48 ± 0.25 s vs. 12.26 ± 0.17 s, $t(5) = 20.16$, $P = 5.55 \times 10^{-6}$, $n = 6$ mice; D2-6-10 test, 18.05 ± 0.79 s, vs. 13.51 ± 0.51 s, $t(5) = 13.75$, $P = 3.65 \times 10^{-5}$, $n = 6$ mice, paired $t$-test) (Figs. 5a–c and 6a–c; Supplementary Fig. 7). These results were significantly seen in each of the six mice tested (D6 test, $P < 0.043$; D4-6-8 test, $P < 0.0014$; D2-6-10 test, $P < 4.19 \times 10^{-6}$, Mann–Whitney $U$-test) (Supplementary Fig. 8). For WT mice ($n = 5$), we confirmed that the waiting time in the blue light trials was not significantly different from that in the yellow light trials in both D6 and D2-6-10 tests (D6 test, 11.62 ± 0.66 s vs. 11.66 ± 0.63 s, $t(4) = 0.90$, $P = 0.42$; D2-6-10 test, 14.61 ± 0.59 s, vs. 14.66 ± 0.70 s, $t(4) = 0.39$, $P = 0.72$, paired $t$-test) (Fig. 6a, c). In D6 and D2-6-10 test, we analyzed WT group data with ChR2 group in a two-way ANOVA. There was a significant main effect of light (two levels within-subject factors; yellow and blue, D6 test, $F(1,9) = 226.75$, $P < 10^{-6}$; D2-6-10 test, $F(1,9) = 139.82$, $P < 10^{-6}$) but no significant main effect of group (two levels between-subject factors; ChR2 and WT, D6 test, $F(1,9) = 0.0028$, $P = 0.96$; D2-6-10 test, $F(1,9) = 1.92$, $P = 0.20$). There was a significant main effect of interaction (light × group, D6 test, $F(1,9) = 259.83$, $P < 10^{-6}$; D2-6-10 test, $F(1,9) = 145.60$, $P < 10^{-6}$). There was a significant simple main effect of light in ChR2 (D6 test, $F(1,9) = 534.62$, $P < 10^{-6}$; D2-6-10 test, $F(1,9) = 313.89$, $P < 10^{-6}$) but no significant simple main effect of light in WT (D6 test, $F(1,9) = 0.52$, $P = 0.49$; D2-6-10 test, $F(1,9) = 0.03$, $P = 0.87$) (Fig. 6a, c).

Among the three delay conditions, the waiting time ratio was largest in the D2-6-10 test (D6 test, 1.12 ± 0.01, $n = 47$ tests; D4-6-8 test, 1.19 ± 0.01 s, $n = 50$ tests; D2-6-10 test, 1.34 ± 0.02 s, $n = 54$ tests) ($H(4) = 110.22$, $P < 10^{-6}$, Kruskal–Wallis test; $P = 8.60 \times 10^{-4}$ for D6 vs. D4-6-8, $P < 10^{-6}$ for D6 vs. D2-6-10, post hoc Bonferroni correction) (Fig. 6e). In each of the six mice tested, the waiting time ratio was the largest in the D2-6-10 test ($P < 0.015$, Mann–Whitney $U$-test) (Supplementary Fig. 9). These results show that serotonin promotes waiting more effectively when mice cannot predict the timing of the reward delivery.

Next, we examined whether the increased waiting time ratio in the D2-6-10 test was due to the introduction of the longest delay (10 s). We introduced a D10 test, in which reward delay was fixed at 10 s with a 75% probability. In the D10 test, waiting time for omission trials with serotonin neuron activation (19.40 ± 0.59 s) was significantly longer than that without serotonin neuron activation (17.55 ± 0.56 s, $t(3) = 13.75$, $P = 8.32 \times 10^{-4}$, $n = 4$ mice, paired $t$-test) (Figs. 5d and 6d; Supplementary Fig. 7).

With regard to waiting time ratio, we performed analysis based on a linear mixed model, taking MI as a random effect. The result of likelihood ratio test between the model with effects of reward-delay condition and the model without this covariate supports the former model ($\chi^2(4) = 133.04$, $P < 10^{-6}$). Further, using the obtained model, we tested difference of mean waiting time ratios between different levels of reward-delay conditions. The mean waiting time ratio of D2-6-10 test was significantly larger than the remainder of the time delay conditions ($Z = 11.0$, $P < 10^{-6}$ for D3 test; $Z = 11.3$, $P < 10^{-6}$ for D6 test; $Z = 10.5$, $P < 10^{-6}$ for D10 test; $Z = 7.91$, $P < 10^{-6}$ for D4-6-8 test). Also, the mean waiting time ratio of D4-6-8 test was significantly large than D6 and D10 tests ($Z = 3.35$, $P = 8.07 \times 10^{-4}$; $Z = 3.50$, $P = 4.64 \times 10^{-4}$,

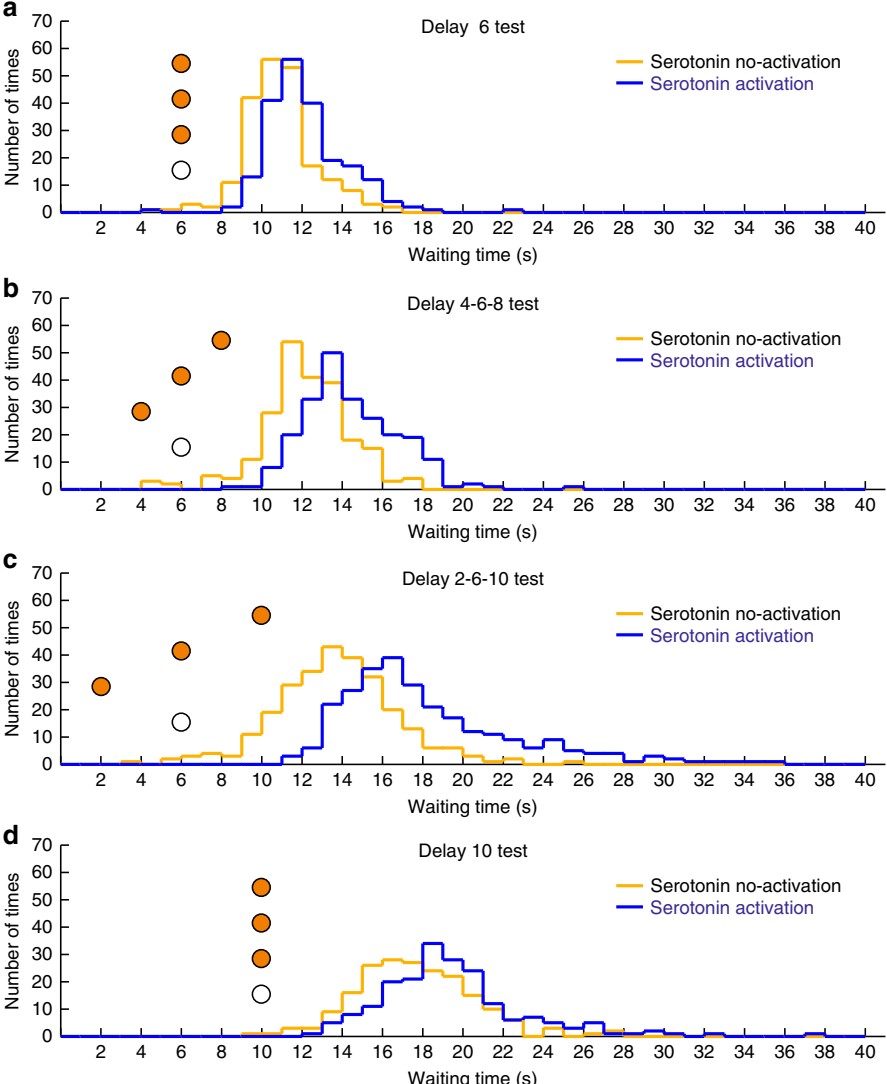

**Fig. 5** Optogenetic activation of DRN serotonin neurons enhances waiting for temporally uncertain rewards. **a** Distribution of waiting time during omission trials in the D6 test. **b** Distribution of waiting time during omission trials in the D4-6-8 test. **c** Distribution of waiting time during omission trials in the D2-6-10 test. **d** Distribution of waiting time during omission trials in the D10 test. Orange circles illustrate the timing and number of food pellets presented in rewarded trials. White circles denote omission trials

respectively). The remainder of differences were not significant (Supplementary Table 2). Subsequently, we tested variability of waiting time among mice. We compared the obtained mixed model with the model including a fixed effect of reward-delay condition, but not a random effect of MI. To evaluate likelihood ratio of two models, we generated 1000 new samples of waiting time ratios by means of a parametric bootstrap method. The variability of waiting time ratio among mice was not significant ($P = 0.602$).

The waiting time ratio in the D6 test was not significantly different from the waiting time ratio in the 75% one-pellet test with a 3 s delay in experiment 1 (D3 test) ($P = 1.00$, post hoc Bonferroni correction) (Fig. 6e). The waiting time ratio in the D10 test ($1.11 \pm 0.01$, $n = 34$ tests) was not significantly different from the waiting time ratios in the D6 test of experiment 2 ($P = 1.00$, post hoc Bonferroni correction) and in the D3 test of experiment 1 ($P = 1.00$, post hoc Bonferroni correction) (Fig. 6e). These results show that timing uncertainty, but not the longest waiting time for future rewards, is critical for enhancing serotonin's effect at increasing waiting times.

**Bayesian decision model of waiting**. Can these effects of serotonin on waiting, depending on the RP and timing uncertainty, be explained in a coherent way? Here we consider the possibility that serotonin signals the prior probability of reward delivery in a Bayesian model of repeated decisions to wait or to quit. In this model, the subject has an internal model of the timing of reward delivery and infers whether the current trial is a reward trial or a no-reward trial. As time goes by without a reward delivery, the likelihood of its being a reward trial diminishes (Fig. 7a, top panel). The posterior probability of a reward follows the same time course scaled by the prior probability for a reward trial (Fig. 7a, middle panel). The expected reward for waiting goes down accordingly and the subject quits waiting as the expected reward for waiting becomes close to that for quitting (zero). The distribution of the time of quitting shifts later as the prior probability of a reward trial increases (Fig. 7a, bottom panel).

If we assume that dorsal raphe serotonin neuron stimulation causes an increase in the estimate of the prior probability when the RP is high, the effect on the waiting time distribution with different RPs (Fig. 2) can be reproduced (Fig. 7b). As the

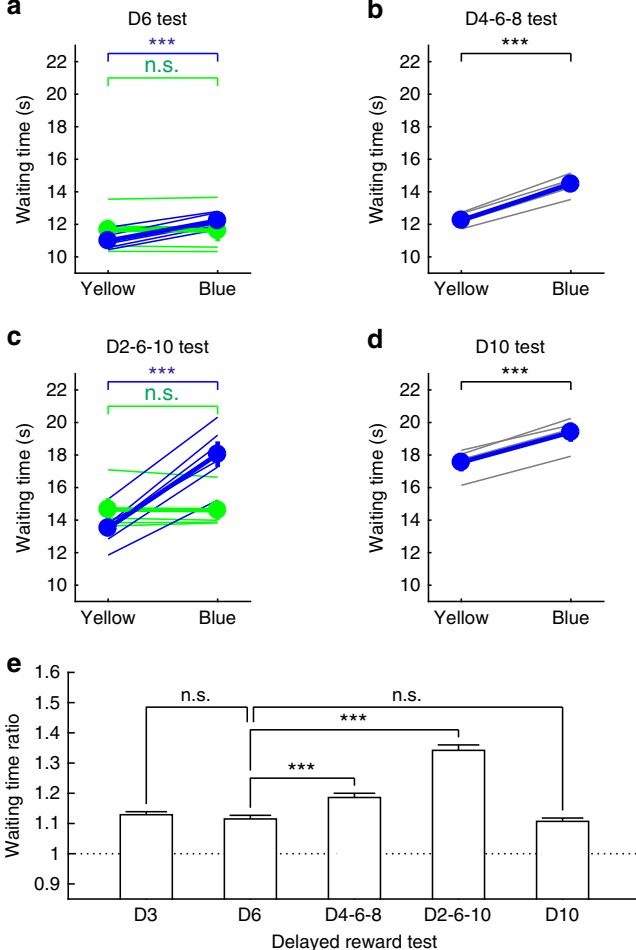

**Fig. 6** The role of serotonin in promoting patience for future rewards with uncertain timing. **a** Average waiting time in serotonin no-activation (yellow) and activation (blue) during the D6 test for individual ChR2-expressing (blue thin lines) and WT (green thin lines) mice and for population of ChR2-expressing (blue line) and WT (green line) mice. **b** Average waiting time in serotonin no-activation (yellow) and activation (blue) during the D4-6-8 test for individual ChR2-expressing mice (gray lines) and for population of mice (blue line). **c** Average waiting time in serotonin no-activation (yellow) and activation (blue) during the D2-6-10 test for individual ChR2-expressing (blue thin lines) and WT (green thin lines) mice and for population of ChR2-expressing (blue line) and WT (green line) mice. **d** Average waiting time in serotonin no-activation (yellow) and activation (blue) during the D10 test for individual ChR2-expressing mice (gray lines) and for population of mice (blue line). ***$P < 0.001$ by paired $t$-test. Error bars represent the s.e.m. In some case, the error bars are too small to be visible. **e** Waiting time ratios in the 75% one-pellet tests in which food pellets were delivered with uncertain timing. The waiting time ratio in the D2-6-10 test was the largest among the five tests. The waiting time ratio in the D6 test was not significantly different from the waiting time ratios in the D3 test and in the D10 test. ***$P < 0.001$ by post hoc Bonferroni correction. Error bars present the s.e.m. n.s. not significant

uncertainty of reward timing increases, the likelihood of a reward trial has a longer tail in the time axis. Accordingly, the same increase in the prior probability causes a larger shift in waiting time distribution (Fig. 7c). This effect approximates the differential effects of serotonin neuron stimulation with different timing uncertainty (Fig. 5).

## Discussion

Through a series of studies, we revealed a causal relationship between dorsal raphe serotonin neuron activation and patience to wait for future rewards[1,2,4,7]. Previous recording studies have shown that DRN neural activity is correlated with levels of behavioral arousal[9], rhythmic motor outputs[10], salient sensory stimuli[11–14], conditioned cues[13–17], rewards[2,13,15–17], reward values and expectation[15–17], punishments[17,18], waiting for delayed rewards[2], and reward omission[13]. Classically, putative serotonin neurons have been identified by broad spikes, slow regular firing, and suppression of 5-HT$_{1A}$ receptor antagonist[2,19,20]. However, it has been difficult to precisely identify serotonergic neurons using these criteria[21–24]. Response diversity in the DRN may reflect non-selective recording of both serotonin and non-serotonin neurons. Using ontogenetic tagging, recent recording studies have demonstrated that serotonin neurons respond to conditioned cure[25,26], reward[3,26], punishment[26], average reward rate[26], and waiting[3]. This response diversity may reflect anatomical, neurochemical, and electrophysiological heterogeneity of serotonergic neurons in the DRN[27]. Nevertheless, 79% of classically identified putative serotonergic neurons[2] and 90% of optogenetically identified serotonergic neurons[3] were tonically activated during waiting for delayed rewards, suggesting that regulating waiting behavior for delayed rewards is a principal function of the serotonin system.

In the current study, we found that optogenetic activation of dorsal raphe serotonergic neurons was not always sufficient to enhance waiting for future rewards. In experiment 1, we found that in the 75% reward test, but not in the 25 or 50% reward tests, optogenetic serotonin activation promoted waiting. These results suggest that a high expectation or confidence in future rewards is necessary for serotonin neural activation to promote waiting and that the interaction of increased serotonin release and the cognitive state of the subject is crucial. Our finding that serotonin neuron activation did not enhance waiting time in the 25 and 50% reward tests also showed that under our stimulation parameters, optogenetic serotonin activation itself did not induce a reinforcing effect to cause prolonged nose poking at the reward site[7,8,25,28,29].

In experiment 2, we found that the effect of serotonin neuron activation on promoting patience was modulated by the variability of timing of reward presentation. Serotonin neuron activation enhanced waiting more effectively when the mice could not predict the timing of the delivery of highly certain rewards. This effect, most prominently observed in D2-6-10 condition, did not simply depend on the average or maximal waiting time because the average waiting time was the same among the D6, D4-6-8, and D2-6-10 conditions and the maximal waiting time was the same between the D10 and D2-6-10 conditions (Fig. 6e). When the timing of reward delivery becomes variable, it becomes more difficult to reject the possibility that the reward may still come. The resulting lower confidence in no reward, or higher subjective probability of reward delivery, might be a reason for the stronger effect of serotonin in facilitating reward-directed behavior.

How does serotonin neuron activation promote patience in waiting? A possible explanation is that serotonin affects the perception of time, such that the same physical time is perceived to be shorter with serotonin neuron stimulation[30]. However, our previous experiment showed that serotonin neuron stimulation during an early phase of waiting does not affect waiting time[7], which is inconsistent with the time perception hypothesis. We previously hypothesized that serotonin controls the temporal discounting parameter in the model-free reinforcement learning framework[31]. While this hypothesis was consistent with many of the recording and manipulation experiments[2,4,7,32], the effects

depending on the RP and timing uncertainty are difficult to explain in terms of a simple temporal discounting paradigm.

Thus, we considered a Bayesian model in which serotonin neuron stimulation affects the prior probability for the present trial to be a reward trial. Our simulation results (Fig. 7) reproduced the critical features of the shifts in waiting time distribution

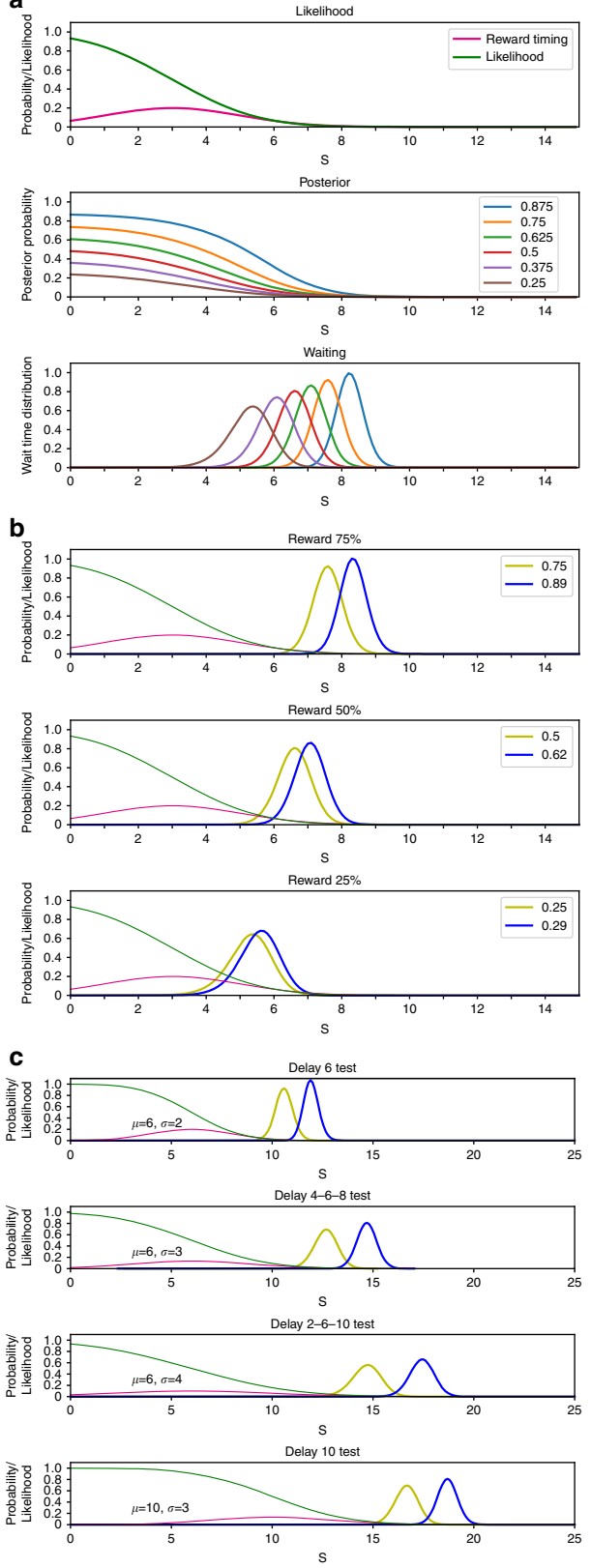

depending on RP and timing uncertainty. The present model is based on several arbitrary assumptions, namely, the internal model of reward timing distribution is Gaussian while the experimental setting is multi-modal, serotonin neuron stimulation causes overestimation of RP especially when the RP is high, and the choice of some free parameters. Nevertheless, this model is consistent with the effect of serotonin on emotional bias toward positive outcomes[33] and a recent report that serotonergic neuron activity keeps track of average reward rate[26], and further points to the possibility of a generalized role of serotonin in arbitrating the trade-off between (negative) sensory evidence and (positive) subjective belief.

Selective serotonin reuptake inhibitors (SSRIs) are widely used to treat psychiatric disorders, especially depression, by increasing the serotonergic tone in the whole brain[34,35]. However, remission rate is 36.8% for citalopram treatment alone[36]. Psychological treatment, such as cognitive behavioral therapy combined with antidepressant therapy, is associated with a higher improvement rate than drug treatment alone[37]. Our finding that activation of serotonin neurons alone is not enough and that it requires a subject's confidence in a positive outcome (i.e., high probability for a future reward) to promote a goal-directed behavior, may explain the combined effect of SSRI treatment and cognitive therapies, which often removes patients' negative biases in future outcomes. The effect of cognitive behavioral therapy is gradual, such that subjects cannot predict a specific time till recovery. Our results in experiment 2 suggest that augmentation of serotonergic tone by SSRI treatment is most effective for enhancing patience for a gradual recovery, and could prevent patients from dropping out. Therefore, SSRI treatment and cognitive behavioral therapy may produce mutually positive effects to realize synergistic therapy.

A recent study showed that inactivation of the orbitofrontal cortex (OFC) disrupts waiting-based confidence reports without affecting decision accuracy[38]. Previous recording studies have also revealed that OFC neurons encode predictions of reward outcomes[39,40]. Optogenetic serotonin activation modulates reward anticipatory responses of OFC neurons[41]. These results suggest that the OFC may produce causal signals for waiting with serotonin neural activation[42]. Optogenetic stimulation of the terminal sites to which DRN serotonin neurons project will clarify the sites where serotonin contributes to enhance patience[43]. Recent rabies virus tracing strategies have yielded a

**Fig. 7** A Bayesian decision making model for waiting reproduces features of effects of reward probability and timing uncertainty on promotion of patience by serotonin. **a** Top panel: the model assumes that the subject has a probabilistic model of reward delivery timing (magenta line), which is assumed to be Gaussian with $\mu = 3$ s and $\sigma = 2$ s in this example. As the time passes without reward delivery, the likelihood of a reward trial diminishes according to the cumulative density function (green line). Middle panel: the posterior probability for a reward trial goes down along with the likelihood, but persists longer if the prior probability for a reward trial is higher. Bottom panel: the timing of quitting is shifted later with a higher prior probability (Methods). **b** We assume that dorsal raphe serotonin neuron stimulation causes an overestimation of the prior probability when the reward probability is higher ($p' = p + p^2 - p^3$ in this example). The yellow and blue lines show the time of quitting without and with increased prior probability, respectively. The effect of serotonin neuron stimulation is largest with a reward probability $p = 0.75$ (top panel; $\mu = 3$ s and $\sigma = 2$ s). **c** With a larger uncertainty $\sigma$ of reward timing, the waiting time distribution shifts later and the effect of serotonin neuron stimulation (increase of prior probability from 0.75 to 0.95 in this example) increases. A shift in the average reward timing (bottom panel; $\mu = 10$ s and $\sigma = 3$ s) does not cause a large increase in waiting time with serotonin neuron stimulation

comprehensive map of afferent inputs to serotonin neurons[44–46]. The combination of serotonergic neural recording with optogenetic manipulation of their afferent inputs will allow us to dissect the afferent inputs, local circuits, and cellular auto-regulatory mechanisms that shape activities of serotonin neurons[47]. These techniques should also allow us to reveal the brain's algorithm for regulation of patience[31].

## Methods

**Animals**. All experimental procedures were performed in accordance with guidelines established by the Okinawa Institute of Science and Technology Experimental Animal Committee. Serotonin neuron-specific ChR2(C128S)-expressing mice were produced by crossing Tph2-tTA mice with tetO-ChR2 (C128S)-EYFP knock-in mice[5,6]. Seven male bigenic and five male WT mice, aged >4 months at the beginning of the behavioral training period, were used in the study. Animals were housed with one mouse per cage at 24 °C on a 12:12 h light: dark cycle (lights on 07:00–19:00 h). Seven bigenic (one for experiment 1 only, one for experiment 2 only, five for both experiments 1 and 2) and five WT animals contributed to the data reported here. Training and test sessions were conducted during the light period 5 days per week. Mice were deprived of food in their home cage and received their daily food ration during the experimental sessions only (~2–3 g per day). Food was freely available during the weekend and removed >15 h before the experimental sessions started. Water was freely available in the home cage.

**Surgery**. After mice had mastered the sequential tone-food waiting task, they were anesthetized with equithesin (3 ml/kg, i.p.), and an optical fiber (400 μm diameter, 0.48 NA, 4 mm length, Doric Lenses) was stereotaxically implanted above the DRN (from bregma: posterior, −4.6 mm; lateral, 0 mm; ventral, −2.6 mm). The optical fiber was fixed to the skull and anchored with dental acrylic and stainless steel screws. Animals were housed individually after surgery and were allowed at least 1 week to recover.

**Reconstruction of optical stimulation sites**. Mice were deeply anesthetized with 100 mg/kg sodium pentobarbital i.p. and were then perfused with 0.9% NaCl, followed by 10% formalin. Their brains were removed and stored in 10% formalin for a minimum of 24 h before being sliced into 60 mm coronal sections. Cresyl violet staining was used to help verify placements of optical fiber tracks (Fig. 1c).

**Behavioral apparatus and training**. A free operant task that we designated as a sequential tone-food waiting task was used. Mice were individually trained and tested in an operant-conditioning box (Med-Associates) measuring 21.6 cm × 17.8 cm × 12.7 cm. The box could be illuminated with a single 2.8 W house light located in the top center of the rear wall. One speaker was positioned in the top right side of the rear wall. Three 2.5 cm square apertures were positioned 2 cm above the floor. The rear stainless steel wall of the chamber contained one aperture defined as the tone site. On the front wall, two apertures defined as the food sites were positioned 7 cm apart. Both apertures on the front wall were connected to a food pellet dispenser that delivered a food pellet (20 mg) to these apertures. In all experiments, only the right food site was used, and the left aperture was covered with an opaque window to prevent nose poking. An infrared photo-beam crossed the entrances of all of the apertures to detect nose poke responses positioned at a depth of 0.5 and 1 cm from the bottom of the aperture. The operant box was illuminated by a house light and was enclosed in a sound-attenuating chamber equipped with a ventilation fan. When the mouse poked its nose through the apertures in the back and front walls, the control infrared photo-beam was interrupted to detect the mouse's responses. The tone site nose poke induced an 8 kHz tone (0.5 s, 85 dB) from the speaker. At the food site, a small food pellet (20 mg) was delivered into the aperture through the food dispenser. All experimental data were recorded with an EPSON personal computer that was connected to the operant box via an interface using MED-PC IV software (Med-Associates).

The beginning of the sequential tone-food waiting task was signaled by turning on the house light, and termination was indicated by turning off the house light. The behavioral instrumental response in this task was for the mouse to hold its nose in a fixed posture in either the tone site aperture while waiting for the conditioned reinforcer tone or the reward site aperture while waiting for the food reward. This task required the mice to perform alternate visits and nose pokes to the tone site and the reward site. The mouse initiated a trial by nose poking in a fixed posture to achieve continuous interruption of the photo-beam at the tone site during a delay period until the tone was presented, signaling that a food reward was available at the reward site. After the tone was presented, the mouse was required to continue nose poking at the reward site during another delay period until the reward was delivered. The delay period that preceded the tone was called the tone delay and that which preceded the food was termed the reward delay. During the initial training period, the tone delay and the reward delay were fixed at 0.2 s.

Two types of error were present in this task: the tone wait error and the reward wait error. The tone wait error and the reward wait error occurred when the mouse

failed to wait for the tone and the food, respectively, during the delay period, by keeping its nose in a fixed posture. After the tone wait error, the mouse could restart the trial until it succeeded in waiting for the tone. A trial ended when the mouse received the food or a food wait error. During a trial, the tone wait error could occur multiple times. By contrast, the reward wait error could occur only one time. Occurrences of tone and reward wait errors were not signaled. Mice could start the next trial at any time after food consumption or after making a reward wait error. Mice were trained daily for a period of 2 h. In 2 weeks or less, mice learned the sequential tone-food waiting task.

**In vivo optical stimulation during the task**. During the test session, an external optical fiber (400 μm diameter, 0.48 NA, Doric Lenses) was coupled to the implanted optical fiber with a zirconia sleeve. The optical fiber was connected to an optic swivel (Doric Lenses) that allowed unrestricted in vivo illumination. The optic swivel was connected to 470 nm blue and 590 nm yellow LEDs (470 nm: 35 mW, 590 nm: 10 mW, Doric Lenses) to generate the blue and yellow light pulses through the optical fiber (960 μm diameter, 0.48 NA, Doric Lenses). Blue and yellow light power intensities at the tip of the optical fiber, as measured by the power meter, were 1.2–2.8 mW and 1.4–1.8 mW, respectively. The LED was controlled by the transistor-transistor-logic pulses generated by a MED-PC IV.

**Experiment 1: effect of reward probability and reward value**. To examine whether reward prediction modulates the effect of serotonin on patience during waiting, we prepared six tests in which the RP and the reward amount were changed (75% reward one-pellet, 75% reward two-pellet, 25% reward one-pellet, 25% reward three-pellet, 50% reward one-pellet, and 50% reward three-pellet tests) (Supplementary Fig. 1). The tone and reward delays were fixed at 0.3 and 3 s, respectively. One test of experiment 1 lasted 3000 s or until the mouse completed 40 trials. The tones in the 75% one-pellet, 75% two-pellet, 25% one-pellet, 25% three-pellet, 50% one-pellet, and 50% three-pellet tests were set at 8 kHz (0.5 s), white noise (0.5 s), 2 kHz (0.25 s) followed by 7 kHz (0.25 s), click (0.5 s), 7 kHz (0.25 s) followed by 2 kHz (0.25 s), and 2.5 kHz (0.5 s), respectively. Removing the nose for >500 ms before the end of the reward-delay period caused a reward wait error, in which no reward was presented. The trials in which serotonin neurons were or were not optogenetically stimulated were named serotonin activation trials or serotonin no-activation trials, respectively (Supplementary Fig. 1). For serotonin activation trials, 0.8 s of blue light was randomly applied for half of the trials at the onset of the nose poke to the reward site following the tone presentation. For serotonin no-activation trials, 0.8 s of yellow light were applied for half of the trials at the onset of the nose poke to the reward site following tone presentation. One trial was ended by applying 1 s of yellow light at the onset of food presentation or the reward wait error (Supplementary Fig. 1).

We executed 75, 25, and 50% reward tests separately. The sequence of 75, 25, and 50% tests was changed for each mouse. During the 75% reward test, 1 or 2 days were used for training in the one-pellet and two-pellet tests and then the recording sessions were started. Each mouse experienced both the one-pellet and two-pellet tests at least once per day. During recording sessions, the order of the one-pellet and two-pellet tests was counterbalanced by daily recording. During both the 25 and 50% reward tests, 1 or 2 days were used for training in the one-pellet and three-pellet tests and then recording sessions were started. Each mouse experienced both one-pellet and three-pellet tests at least once per day. During the recording sessions, the order of the one-pellet and three-pellet tests was counterbalanced by daily recording.

**Experiment 2: effect of reward timing uncertainty**. To examine whether the timing of presentation of an expected reward influences promotion of patience by serotonin, we prepared four delayed reward tests with 75% RP, in which the timing of reward delivery was fixed at: (i) the reward delay was fixed at 6 s (D6 test) (Supplementary Fig. 6a); (ii) the reward delay was randomly set to 4, 6, or 8 s (D4-6-8 test) (Supplementary Fig. 6b); (iii) the reward delay was randomly set to 2, 6, or 10 s (D2-6-10 test) (Supplementary Fig. 6c); and (iv) the reward delay was fixed at 10 s (D10 test). One test of experiment 2 lasted 3000 s or until the mouse completed 40 trials. The tone was 0.5 s at 8 kHz and was fixed through four reward-delay conditions. Removing the nose for >500 ms before the end of the reward-delay period caused a reward wait error, in which no reward was presented. Light stimulation patterns during the serotonin activation and serotonin no-activation trials were the same as in experiment 1. In the D4-6-8 and D2-6-10 tests, the eight trial patterns (two light conditions multiplied by four delay lengths) were randomly selected without repetition until all items were selected, and then this selection was repeated five times. In the D6 and D10 tests, eight trials (three fixed delay with serotonin activation, one omission with serotonin activation, three fixed delay without serotonin activation, and one omission without serotonin activation) were randomly selected without repetition until all items were selected, and then this selection was repeated five times.

We executed the D6, D4-6-8, D2-6-10, and D10 test sessions in this order. In each reward-delay test session, the first day was a training session followed by 3 or 4 days of recording sessions. The 1-day recording sessions consisted of at least one reward-delay test. For two mice, D4-6-8 and D6 test sessions were further executed in this order after D2-6-10 test session (one mouse) or D10 test session (one mouse). Since in both D6 and D4-6-8 test sessions, waiting time in omission trials

did not differ significantly between first and second sessions, data from first and second sessions were merged for analysis (in the D6 test, $P > 0.10$ with serotonin activation, $P > 0.10$ without serotonin activation, Mann–Whitney $U$-test; in the D4-6-8 test, $P > 0.79$ with serotonin activation, $P > 0.13$ without serotonin activation, Mann–Whitney $U$-test).

**Data analysis**. No statistical tests were used to determine sample size, but our sample sizes were similar to those employed in our previous study[7]. To examine how serotonin neuron activation promotes waiting for delayed rewards, we focused on waiting time during omission trials. To quantify effectiveness of serotonin neuron activation at promoting waiting time during omission trials, we calculated the waiting time ratio (waiting time with serotonin neuron activation/waiting time without serotonin neuron activation) for each test. Statistically significant differences (waiting time or waiting time ratio) between two groups were assessed by Mann–Whitney $U$-test. To compare waiting time in serotonin activation and in serotonin no-activation by within animal averages, we used paired $t$-test. For analysis of ChR2-expressing group (ChR2) data and control group (WT) data, two-way ANOVA using light effect (two levels; yellow and blue) as within-subject factors and group effect (two levels; ChR2 and WT) as between-subject factors were used. The normality of data for paired $t$-test and two-way ANOVA were assessed by Shapiro–Wilk test. We have checked a homogeneity of variance of the waiting time ratio data in experiments 1 and 2. Since data did not satisfy homogeneity of variance in both experiment, non-parametric statistical tests were used. To examine the main effect of RP (three level; 75, 50, and 25%) and that of expected reward value (four levels; 0.25, 0.5, 0.75 and 1.5 EPs per trial) on promoting waiting time, Scheirer–Ray–Hare test, which is non-parametric method equivalent to two-way ANOVA, followed by the Bonferroni correction for multiple comparisons was used for analysis of the waiting time ratio. A linear mixed model analysis was performed, taking the waiting time ratio ($Y$) as a dependent variable, RP, and EP as independent variables with fixed effect, and MI as an independent variable with random effect. We fitted the model to data using R package {lme4} with the formula $Y$ = RP + EP + (1|MI). To test difference of means, we used $Z$-value instead of $t$-value because the degree of freedom of $t$-value is not readily available for an unbalanced mixed model. Further, to test whether variance of mice is zero, it is not appropriate to use a $\chi^2$-test because the null hypothesis is located in the end of domain of variance. As a bail-out method, we used a parametric bootstrap. Kruskal–wallis test followed by Bonferroni correction for multiple comparisons was used for analysis of the waiting time ratio in experiment 2. In Bonferroni correction for multiple comparisons, $P$-values of pairwise Mann–Whitney $U$-tests were multiplied by $m$, where $m$ was the number of pairwise Mann–Whitney $U$-tests. Statistically significant differences were achieved when $P$-value $\times$ $m < 0.05$. $m$ was 15 and 10 in Scheirer–Ray–Hare test and Kruskal–wallis test, respectively. Data collection and analysis were not performed blind during the experiment, and no randomization was used. In a very small number of omission trials, mice removed the nose from the reward site within 1.5 s (in the 75% one-pellet test, 2 for serotonin activation trial and 2 for serotonin no-activation trial; in the 50% three-pellet test, 3 for serotonin activation trial, and 4 for serotonin no-activation trial; in the 50% one-pellet test, 1 for serotonin activation trial; in the 25% three-pellet test, 4 for serotonin activation trial and 2 for serotonin no-activation trial; in the 25% one-pellet test, 1 for serotonin activation trial and 1 for serotonin no-activation trial; in the D10 test, two for serotonin no-activation trial). These data were excluded from the analysis. Statistical analyses were performed using SPSS, Matlab (MathWorks), and R.

**Bayesian decision model of waiting**. Each trial had a hidden state $X$ = {reward, no-reward}, and for a reward trial, the timing of reward delivery was given by a Gaussian distribution $N(t; \mu, \sigma^2)$. Given an observation that a reward had not been delivered by time $t$, the likelihood for a reward trial was $1 - f(t; \mu, \sigma^2)$, where $f$ is the cumulative Gaussian density function, whereas the likelihood for a no-reward trial was one. The posterior probability for a reward trial, given observation of no reward by time $t$ is

$$P(\text{reward}|t) = P(\text{reward}) \times (1 - f(t; \mu, \sigma^2))/[P(\text{reward}) \times (1 - f(t; \mu, \sigma^2)) + P(\text{no reward})],$$

where $P(\text{reward})$ and $P(\text{no reward})$ are prior probabilities of reward and no-reward trials.

The expected reward to keep waiting was $V(\text{wait}|t) = P(\text{reward}|t)$ for a unit of reward, while the expected reward for quitting was $V(\text{quit}|t) = 0$ as no reward is obtained by quitting. By assuming a softmax action selection, the choice probability to keep waiting at time $t$ is

$$P(\text{wait}|t) = 1/(1 + \exp[-\beta \times P(\text{reward}|t)]),$$

where $\beta$ is the inverse temperature parameter regulating the stochasticity of choice. The distribution of the time of quitting $P_{\text{quit}}(t)$ is given by sequential decisions:

$$P_{\text{wait}}(0) = 1,$$
$$P_{\text{wait}}(t) = P_{\text{wait}}(t - \tau) \times P(\text{wait}|t),$$
$$P_{\text{quit}}(t) = P_{\text{wait}}(t - \tau) \times (1 - P(\text{wait}|t)),$$

where $P_{\text{wait}}(t)$ is the probability of continuing to wait until time $t$ and $\tau$ is the interval of repeated decision to wait or to quit. In Fig. 7, we used parameters $\tau = 0.1$ s and $\beta = 50$. The code of the Bayesian waiting decision model was written in Python.

**Code availability**. The code used to generate the results that are reported in this study are available from the corresponding author to responsible request.

**Data availability**. Data from the experiments presented in this study are available from the corresponding author to responsible request.

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

## Acknowledgements

This work was partially supported by a JSPS KAKENHI Grant-in-Aid for Young Scientists (B) 24730643 (to K.W.M.), "Integrated research on neuropsychiatric disorders," performed under the Strategic Research Program for Brain Sciences by the Ministry of Education, Culture, Sports, Science, and Technology of Japan (to K.M., K.W.M., and K.D.), a Grant-in-Aid for Scientific Research on Innovative Areas: Prediction and Decision Making 26120728 (to K.M.) and 23120007 (to K.D.), and a Grant-in-Aid for Scientific Research on Innovative Areas: Elucidation of the Mathematical Basis and Neural Mechanisms of Multi-layer Representation Learning 16H06563 (to K.D.) We thank Aki Takahashi for breeding the mice and for providing the Tph2-tTA::tetO-ChR2(C128S)-EYFP knock-in mice. We also thank members of the Neural Computation Unit for their helpful comments and discussion.

## Author contributions

K.M., K.W.M., and K.D. designed the research. K.M. and K.W.M. performed the experiments. K.M. and K.W.M. and T.T. analyzed the data. K.M., K.W.M., and K.D. discussed the results and wrote the manuscript. A.Y. and K.F.T. generated the Tph2-tTA::tetO-ChR2(C128S)-EYFP knock-in mice. All authors edited the manuscript.

## Additional information

**Competing interests:** The authors declare no competing interests.

