## [Peer Review File · Nature Communications]

Reviewers' comments:

Reviewer #1 (Remarks to the Author):

Miyazaki, Miyazaki, et al. manipulated the activity of serotonin neurons in mice engaged in a task in which they waited for reward. This subject is important and timely, as the functions of serotonin neurons are unclear. Unfortunately, there is a key control experiment missing, the data do not support the strong conclusions, and several points are confusing.

Major comments:

1. The manuscript is missing a critical control experiment: AAV-EYFP injections into transgenic mice (or tetO-EYFP mice, if available), together with the stimulation protocol. This would rule out a trivial explanation of the data: that mice interpreted the first yellow light on "no stimulation trials" as a trial end cue, especially because it is the same yellow light that is intended to be a trial end cue shortly thereafter. This alternative interpretation could explain how the largest effects occurred on high probability and high timing uncertainty trials.
2. The largest effects occur when reward probability is high (i.e., low uncertainty) and reward timing is variable (i.e., high uncertainty). What does this interaction between uncertainty and probability/timing mean? In the Discussion, the authors claim that a necessary condition for serotonin neuron activation to extend waiting for reward is "confidence in a positive outcome" (line 171). This is very speculative. What is the evidence that confidence about positive outcomes increases as uncertainty about timing increases? Any strong claim here would require experimental support in this manuscript.
3. The claim that serotonin stimulation promotes patience is strong. Could the effects also be explained by "behavioral inhibition"? To address this, it would be useful to use reaction times, as Fonseca et al. (2015) did. Similarly, it is unclear why the same behavioral measurement is described as "waiting time" in one condition (no stimulation) but "promotion of patience" in another condition (stimulation) (lines 76-78, 93-95).
4. The results here are difficult to understand in the context of recent recordings from unidentified and identified neurons in dorsal raphe (Nakamura et al., J Neurosci, 2008; Ranade and Mainen, J Neurophysiol, 2009; Li et al., J Neurosci, 2013; Cohen et al., eLife, 2015). These studies should be cited and discussed, particularly in light of the heterogeneity of dorsal raphe neuronal firing rates.
5. To ensure that there is no bias between trial types, it is necessary to see that there was no difference in 1) reward errors and 2) time between tone nose poke and reward site nose poke across days. Given that the a session could end after animals completed 40 trials, animals could adopt a strategy in which they "throw out" the session on a low probability day (i.e., higher reward errors on 25% probability trials compared to 75% probability trials). Was food intake and weight stable across days? Additionally, given that a session could also end after 3000 s, animals could show differences in time approaching the reward port (such that they take longer on a 25% probability day) if they are less motivated due to lower payoff. Also, given that mice had ad lib access to food on weekends, was behavior slower and less accurate on Mondays? Please show performance across training.
6. Please ensure that all statistical tests use corrections for multiple comparisons. Planned contrasts would be more appropriate here than a series of nonparametric two-way tests.

Minor comments:

1. The histograms should not be plotted as interleaving bars, given that the abscissa is a continuous random variable. It makes the blue bars appear artificially shifted relative to the yellow ones.
2. Given the claim that timing uncertainty is needed for enhancing serotonin's effect on increasing waiting times, it would be interesting to know if the number of uncertain elements (e.g., both the D4-6-8 and D2-6-10 have three wait times) or the variance (e.g., the D2-6-10 has a larger spread of wait times) contributes to the effect.
3. Why were experiments performed during the light cycle for mice?
4. The last sentence of the abstract is very speculative and would be more appropriate in the Discussion, if at all.
5. The text and figures would benefit from judicious editing.
6. Ref. 3 showed experiments in mice, not rats (line 36).

Reviewer #2 (Remarks to the Author):

In this paper, the authors show that the probability and timing uncertainty of future rewards affects the role of dorsal raphe serotonin neurons in promoting patience. There are two key results: 1. Optogenetic activation of serotonin neurons during a reward seeking task extends waiting time if the reward is relatively likely (75% probability of receiving a reward), but has no effect if the reward is less probable (50% or 25% chance); 2. Activation of serotonin neurons extends waiting time if the timing of the reward is uncertain, and this effect is bigger if there is a greater degree of uncertainty. This study is well executed and written, and particularly timely, given the recent surge of papers using optical and genetic techniques to probe the functional role of serotonin in mediating motivated behavior. It is likely to be of broad interest.

I really have no major points. This is a nice, straightforward extension of the authors' previous study (Current Biology 2014), is cleanly designed, utilizes a mouse model that has been validated, and there is a robust behavioral effect. It would be possible to ask for other causal/observational components (inhibition of serotonin neurons, recording of serotonin neurons during this task), but I think that this is unnecessary. It would be nice for the field to have immediate access to these results.

Reviewer #3 (Remarks to the Author):

This is a nicely written manuscript. The results appear to add details to the conditions under which artificial activation of the dorsal raphe serotonin neurons is able to modify the time animals wait for delayed rewards.

While of potentially of interest, the additional parameterisation of the stimulation effects provided by this study would represent incremental build on the prior work rather than a major new direction. Further, there is not a great deal specific that can be inferred from the results in terms of specific theoretical advances in understanding. The results lead to speculations in the discussion, but these

purely behavioural studies cannot directly address these questions, which will require additional studies using a range of other methods, as mentioned in the manuscript.

Another issue is that the stimulation presumably hyperactivates the 5-HT system both by adding additional spiking activity over and above that which would normally be present, and much more synchronously than normal. Changes in behaviour could result from exaggeration of normal 5-HT activity (which seems to be the preferred interpretation) but could also reflect a functional abnormality in downstream targets due to excess 5-HT levels. Even if the former is the case, does such a gain of function approach tell us much directly about the normal role of fluctuations in 5-HT neuron activity within the normal dynamic range?

Technical points:

Is the statistical analysis appropriate? It appears to consist of very many pairwise contrasts. While these use the non-parametric MW-U test, sensible for ratio data, there does not seem to be any correction for multiple comparisons. Also, to make the direct comparisons between different values would seem to require including the various values into a single analysis, such as 2-way ANOVA (or non-parametric equivalent if required) or mixed-model approaches.

Further, it appears that n is total number of trials across all animals, rather than number of animals. Is this appropriate? In any case, this would explain the incredibly small p values – and the rather odd fact that in some cases whereas paired contrasts had extremely low p values, what appear to be the matching ratio tests addressing the same effects (which are probably the more appropriate approach, and it seems unnecessary to have both) were not significant.

Has the behaviour/neurophysiology of the CRE animal been tested compared to the base strain, to ensure the abnormal protein expression has not shifted baseline 5-HT neuron activity?

For 5/7 days animals only received food in the task context and the "reward" was standard chow. Obtaining the food was thus a homeostatic necessity. Could this confound interpretation?

Minor points.

"Patience"; is used as a form of shorthand as if it were synonymous with "enhanced waiting duration", but this is not really the case; patience is a human trait with emotional overtones, relating to the ability to tolerate delay without feelings of annoyance. Thus we can wait, but may do so impatiently. Changes in median waiting time might be a proxy measure for the effects of an underlying emotional state, but we cannot know the inner emotional state of the animals.

The Figure legends for graphs include all the exact mean and SEM values. Unless required by journal style, this seems unnecessary overkill; the Figures alone provide the necessary information, it is not clear that the exact values to two decimal places add anything.

On the other hand, details of the statistics such as U values are not included.

Responses to reviewers' comments

We are grateful to the reviewers for their critical comments and useful suggestions, which have helped us improve our paper. As indicated in the responses that follow, we have taken all these comments and suggestions into account in the revised version of our manuscript.

Reviewer #1

Major comments:

Comment #1

Miyazaki, Miyazaki, et al. manipulated the activity of serotonin neurons in mice engaged in a task in which they waited for reward. This subject is important and timely, as the functions of serotonin neurons are unclear. Unfortunately, there is a key control experiment missing, the data do not support the strong conclusions, and several points are confusing.

The manuscript is missing a critical control experiment: AAV-EYFP injections into transgenic mice (or tetO-EYFP mice, if available), together with the stimulation protocol. This would rule out a trivial explanation of the data: that mice interpreted the first yellow light on "no stimulation trials" as a trial end cue, especially because it is the same yellow light that is intended to be a trial end cue shortly thereafter. This alternative interpretation could explain how the largest effects occurred on high probability and high timing uncertainty trials.

In our previous paper (Miyazaki et al., 2014), we used wild-type mice, instead of AAV-EYFP injected mice or tetO-EYFP mice, to show that blue or yellow light stimulation did not affect waiting time. In the revised manuscript, we have mentioned this earlier result in the Methods regarding Experiment 1 (page 18, lines 17-20).

Comment #2

2. The largest effects occur when reward probability is high (i.e., low uncertainty) and reward timing is variable (i.e., high uncertainty). What does this interaction between uncertainty and probability/timing mean? In the Discussion, the authors claim that a necessary condition for serotonin neuron activation to extend waiting for reward is "confidence in a positive outcome" (line 171). This is very speculative. What is the

evidence that confidence about positive outcomes increases as uncertainty about timing increases? Any strong claim here would require experimental support in this manuscript. What we meant was that with higher uncertainty about timing, judging that the reward is not coming becomes more difficult. In order to demonstrate this link between confidence and timing uncertainty, we now present a Bayesian decision model for waiting. This model assumes that an animal has an internal model of reward timing distribution (red line in Fig.7a top panel) and combines the likelihood of the trial's being rewarded and the prior probability of reward trial to evaluate the posterior probability of a reward. The likelihood of a reward trial drops to zero with the passage of time (green line in Fig.7a top panel), but the posterior probability persists longer if the prior probability for a reward trial is higher (Fig.7a middle panel). Accordingly, the distribution of the timing of quitting is shifted later (Fig.7a bottom panel). If we assume that dorsal raphe serotonin neuron stimulation causes an increase in the estimate of the prior probability when reward probability is high, the effects on the waiting time distribution with different reward probabilities (Fig. 2) can be reproduced (Fig. 7b). As the uncertainty of reward timing is increased, the reward likelihood decreases more gradually. Accordingly, the same increase in the prior probability causes a larger shift in waiting time distribution (Fig. 7c). This effect approximates the differential effects of serotonin neuron stimulation with different timing uncertainty (Fig. 5).

We used the phrase “confidence in a positive outcome” to mean “high prior probability for a trial to be rewarded” in our Bayesian decision model. We describe the detail of the Bayesian decision making model in a new section in the Results (page 8, lines 15 to page 9, line 12), the Discussion (page 11, lines 7 to page 12, line 4), and the Methods (page 21, lines 14 to page 22, line 13) of the revised manuscript.

Comment #3

3. The claim that serotonin stimulation promotes patience is strong. Could the effects also be explained by "behavioral inhibition"? To address this, it would be useful to use reaction times, as Fonseca et al. (2015) did. Similarly, it is unclear why the same behavioral measurement is described as "waiting time" in one condition (no stimulation) but "promotion of patience" in another condition (stimulation) (lines 76-78, 93-95).

In our previous paper (Miyazaki et al., 2014), we showed that blue light stimulation did not induce behavioral inhibition in experiments 5, 12, and 13. These results suggest that prolonged waiting time by serotonin activation could not be explained by simple behavioral inhibition.

We appreciate the comment on ambiguous description of the same behavioral

measurement. We have unified the same behavioral measurement with the term “waiting time” in both no stimulation and with stimulation conditions in the revised manuscript.

Comment #4

4. The results here are difficult to understand in the context of recent recordings from unidentified and identified neurons in dorsal raphe (Nakamura et al., J Neurosci, 2008; Ranade and Mainen, J Neurophysiol, 2009; Li et al., J Neurosci, 2013; Cohen et al., eLife, 2015). These studies should be cited and discussed, particularly in light of the heterogeneity of dorsal raphe neuronal firing rates.

We have cited previous unit recording studies in the DRN and have discussed differences between this study and previous recording studies in the Discussion of the revised manuscript (page 9, line 16 to page 10, line 8).

Comment #5

5. To ensure that there is no bias between trial types, it is necessary to see that there was no difference in 1) reward errors and 2) time between tone nose poke and reward site nose poke across days. Given that the a session could end after animals completed 40 trials, animals could adopt a strategy in which they "throw out" the session on a low probability day (i.e., higher reward errors on 25% probability trials compared to 75% probability trials). Was food intake and weight stable across days? Additionally, given that a session could also end after 3000 s, animals could show differences in time approaching the reward port (such that they take longer on a 25% probability day) if they are less motivated due to lower payoff. Also, given that mice had ad lib access to food on weekends, was behavior slower and less accurate on Mondays? Please show performance across training.

We added detailed information about food deprivation to the Methods of the revised manuscript. Food was freely available during the weekend and was removed more than 15 hours before the experimental sessions started. In Experiment 1, we executed the 75%, 50% and 25% reward tests separately. The order of the 75%, 50% and 25% tests was changed for each mouse. During each probability reward test, one or two days were used for training in the big reward and small reward tests. Training usually started on Monday. After training, we started daily recording sessions up to four consecutive days in a week. Even in the 25% one-pellet test, for each tested mouse we did not “throw out” low-reward sessions. There was no significant main effect of the number of wait errors during four consecutive recording sessions (four level; day 1, day 2, day 3, and day 4,

$F(3,3312) = 1.14, P = 0.30$, two-way ANOVA). There was no significant main effect of tests (two level; 75% test and 25% test, $F(1,3312) = 0.48, P = 0.49$) and no significant interaction between recording sessions and tests (sessions \times tests, $F(3, 3312) = 0.84, P = 0.47$). The number of wait errors did not significantly change across daily recording sessions ($F(3,77) = 0.48, P = 0.70$, two-way ANOVA). There was no significant main effect of tests ($F(1,77) = 2.42, P = 0.12$, two-way ANOVA) and no significant interaction between recording sessions and tests ($F(3, 77) = 1.30, P = 0.28$). Although we did not measure daily mouse weights during recording sessions, there was no significant main effect of the number of food intakes during four consecutive recording sessions ($F(3,36) = 0.57, P = 0.64$, two-way ANOVA). There was no significant main effect of tests ($F(1,36) = 0.55, P = 0.46$) and no significant interaction between recording sessions and tests ($F(3, 36) = 0.49, P = 0.69$). These results showed that task performances of mice were stable in both the 75% one-pellet test and the 25% one-pellet test.

Comment #6

6. Please ensure that all statistical tests use corrections for multiple comparisons. Planned contrasts would be more appropriate here than a series of nonparametric two-way tests. We have reanalyzed the waiting time ratio data in Experiment 1 using two-way ANOVA followed by post hoc Bonferroni test for multiple comparisons. There was a significant main effect of the reward probability (three level; 75%, 50%, and 25%, $F(2,200) = 53.04, P < 10^{-6}$, two-way ANOVA), but no significant main effect of the expected reward value (four level; 0.25, 0.5, 0.75, and 1.5, expected pellets per trial, $F(3,200) = 0.21, P = 0.89$).

We have also reanalyzed the waiting time ratio data in Experiment 2 by one-way ANOVA followed by post hoc Bonferroni test for multiple comparisons.

Minor comments:

Comment #7

1. The histograms should not be plotted as interleaving bars, given that the abscissa is a continuous random variable. It makes the blue bars appear artificially shifted relative to the yellow ones.

We appreciate this kind advice. We have replotted figures 2 and 5 as line graphs to prevent a seeming artificial shift of the waiting time by blue light stimulation.

Comment #8

2. Given the claim that timing uncertainty is needed for enhancing serotonin's effect on increasing waiting times, it would be interesting to know if the number of uncertain elements (e.g., both the D4-6-8 and D2-6-10 have three wait times) or the variance (e.g., the D2-6-10 has a larger spread of wait times) contributes to the effect.

The Bayesian decision model predicts that the effect of serotonin activation on promoting waiting increases when the internal model of reward timing distribution becomes broad (e.g. large variance). To examine how the number of uncertain elements or the variance contributes to serotonin's effect would further strengthen our result, but such an experiment is kept as a future option.

Comment #9

3. Why were experiments performed during the light cycle for mice?

In our behavioral task, turning on the room light means task start and light is continuously on during the task. From this reason, we performed experiments during the light cycle for mice.

Comment #10

The last sentence of the abstract is very speculative and would be more appropriate in the Discussion, if at all.

We appreciate this suggestion and have revised the last three sentences of the abstract (page 2, lines 9-18).

Comment #11

The text and figures would benefit from judicious editing.

We appreciate your suggestion. The manuscript had been sent to a proofreading company to check for grammatical errors. We have now sent it to OIST's technical editor, who has edited it judiciously.

Comment #12

Ref. 3 showed experiments in mice, not rats (line 36).

We appreciate this correction. We have revised the text (page 3, line 5).

References:

Miyazaki, K. W *et al.* Optogenetic activation of dorsal raphe serotonin neurons enhances patience for future rewards. *Curr. Biol.* **24**, 2033-2040 (2014).

Reviewer #2

Comment #1

In this paper, the authors show that the probability and timing uncertainty of future rewards affects the role of dorsal raphe serotonin neurons in promoting patience. There are two key results: 1. Optogenetic activation of serotonin neurons during a reward seeking task extends waiting time if the reward is relatively likely (75% probability of receiving a reward), but has no effect if the reward is less probable (50% or 25% chance); 2. Activation of serotonin neurons extends waiting time if the timing of the reward is uncertain, and this effect is bigger if there is a greater degree of uncertainty. This study is well executed and written, and particularly timely, given the recent surge of papers using optical and genetic techniques to probe the functional role of serotonin in mediating motivated behavior. It is likely to be of broad interest.

I really have no major points. This is a nice, straightforward extension of the authors' previous study (Current Biology 2014), is cleanly designed, utilizes a mouse model that has been validated, and there is a robust behavioral effect. It would be possible to ask for other causal/observational components (inhibition of serotonin neurons, recording of serotonin neurons during this task), but I think that this is unnecessary. It would be nice for the field to have immediate access to these results.

We appreciate these encouraging comments.

Reviewer #3

Comment #1

This is a nicely written manuscript. The results appear to add details to the conditions under which artificial activation of the dorsal raphe serotonin neurons is able to modify the time animals wait for delayed rewards.

While of potentially of interest, the additional parameterisation of the stimulation effects provided by this study would represent incremental build on the prior work rather than a major new direction. Further, there is not a great deal specific that can be inferred from the results in terms of specific theoretical advances in understanding. The results lead to speculations in the discussion, but these purely behavioural studies cannot directly address these questions, which will require additional studies using a range of other methods, as mentioned in the manuscript.

In order to link our behavioral findings to a theoretical framework, we now present a Bayesian decision model for waiting. Please see our response to Comment #2 of Reviewer #1.

Comment #2

Another issue is that the stimulation presumably hyperactivates the 5-HT system both by adding additional spiking activity over and above that which would normally be present, and much more synchronously than normal. Changes in behaviour could result from exaggeration of normal 5-HT activity (which seems to be the preferred interpretation) but could also reflect a functional abnormality in downstream targets due to excess 5-HT levels. Even if the former is the case, does such a gain of function approach tell us much directly about the normal role of fluctuations in 5-HT neuron activity within the normal dynamic range?

We do not expect that light stimulation during waiting hyperactivated the 5-HT system, for three reasons. First, in this study we used step-function opsin ChR2(C128S), which induced a continuous increase of the inward current (Berndt et al., 2009), which, in turn, induces asynchronous activation of serotonin neurons, unlike pulse-train stimulation of conventional ChR2s, which induces synchronized firing of serotonin neurons (e.g., Liu, 2014, Fig.4b). Second, in our previous paper (Miyazaki et al., 2014), we showed by *in vitro* experiments that the firing rate was about 5 to 6 Hz when the step-opsin was fully activated (Fig. 1e). These increases in the firing rate are comparable to those (about 4 to 5 Hz) when the rats were waiting for delayed rewards (Miyazaki et al., 2011,

Supplemental Fig. S6). Finally, we showed by *in vivo* microdialysis that optogenetic activation of dorsal raphe serotonin neurons increased serotonin efflux in the mPFC on the order of 60-70% (Miyazaki et al., 2014, Fig. 1j). These increases in serotonin efflux are comparable to those in the mPFC (about 50%) when rats were performing the long delayed reward task (Miyazaki et al., 2012, Fig. 2).

Technical points:

Comment #3

Is the statistical analysis appropriate? It appears to consist of very many pairwise contrasts. While these use the non-parametric MW-U test, sensible for ratio data, there does not seem to be any correction for multiple comparisons. Also, to make the direct comparisons between different values would seem to require including the various values into a single analysis, such as 2-way ANOVA (or non-parametric equivalent if required) or mixed-model approaches.

According to this suggestion, we applied 2-way ANOVA in our revised manuscript. Please see our response to Comment #6 from Reviewer #1.

Comment #4

Further, it appears that n is total number of trials across all animals, rather than number of animals. Is this appropriate? In any case, this would explain the incredibly small p values – and the rather odd fact that in some cases whereas paired contrasts had extremely low p values, what appear to be the matching ratio tests addressing the same effects (which are probably the more appropriate approach, and it seems unnecessary to have both) were not significant

The advantage of optogenetic methods compared with pharmacological methods is that animal behaviors can be analyzed on a trial-by-trial basis. In our study, there is no need to separate the serotonin activation group and serotonin no-activation group. We found that optogenetic stimulation of serotonin neurons during waiting for delayed rewards dynamically changed during one test in which there were 40 trials, half of which were randomly selected for optogenetic stimulation. We also found that serotonin's effect on promoting waiting time was consistent within each tested mouse. From these results, we analyzed waiting time during omission trials by the total number of trials across all animals, rather than the number of animals.

Comment #5

Has the behaviour/neurophysiology of the CRE animal been tested compared to the base

strain, to ensure the abnormal protein expression has not shifted baseline 5-HT neuron activity?

In our previous paper (Miyazaki et al., 2014), we showed using an in-slice experiment that the baseline firing rate was ~1-3 Hz (Fig. 1e and 1h). This baseline firing rate is comparable to that (~1-2 Hz) during *in vivo* unit recording (Miyazaki et al., 2011, Fig. 3f).

Comment #6

For 5/7 days animals only received food in the task context and the "reward" was standard chow. Obtaining the food was thus a homeostatic necessity. Could this confound interpretation?

As reported in our response to Comment #5 of Reviewer #1, animal behavior did not significantly change during days after the weekend, when food was freely available. Thus, we expect that food restriction during weekdays did not cause a significant effect on reward-oriented behaviors or brain functions.

Minor points:

Comment #7

"Patience"; is used as a form of shorthand as if it were synonymous with "enhanced waiting duration", but this is not really the case; patience is a human trait with emotional overtones, relating to the ability to tolerate delay without feelings of annoyance. Thus we can wait, but may do so impatiently. Changes in median waiting time might be a proxy measure for the effects of an underlying emotional state, but we cannot know the inner emotional state of the animals.

We appreciate this comment regarding the ambiguous expression about patience. We replaced the word "patience" to "waiting" in the technical description of our experiment, while we still use the word in the Introduction and Discussion for motivation and interpretation of the experiment.

Comment #8

The Figure legends for graphs include all the exact mean and SEM values. Unless required by journal style, this seems unnecessary overkill; the Figures alone provide the necessary information, it is not clear that the exact values to two decimal places add anything.

We appreciate this suggestion and have removed exact mean and SEM values of waiting time and waiting time ratio in the figure legends.

Comment #9

On the other hand, details of the statistics such as U values are not included.

We have provided U values in addition to p-values in the revised manuscript.

References:

Barndt, A., Yizhar, O., Gunaydin, L.A., Hegemann, P. & Deisseroth, K. Bi-stable neural state switches. *Nat. Neurosci.* **12**, 229-234 (2009).

Liu, Z *et al.* Dorsal raphe neurons signal reward through 5-HT and Glutamate. *Neuron* **81**, 1360-1374 (2014).

Miyazaki, K., Miyazaki, K. W. & Doya, K. Activation of dorsal raphe serotonin neurons underlies waiting for delayed rewards. *J. Neurosci.* **31**, 469-479 (2011).

Miyazaki, K.W., Miyazaki, K., and Doya, K. (2012). Activation of dorsal raphe serotonin neurons is necessary for waiting for delayed rewards. *J. Neurosci.* **32**, 10451-10457.

Miyazaki, K. W *et al.* Optogenetic activation of dorsal raphe serotonin neurons enhances patience for future rewards. *Curr. Biol.* **24**, 2033-2040 (2014).

Reviewers' comments:

Reviewer #1 (Remarks to the Author):

The manuscript is improved, but the authors did not address my main concern in the first review.

1. In my previous comment (1), I asked whether mice could have interpreted the first yellow light on "serotonin no-activation" trials as a trial-end cue, because yellow light was also used at the end of all trials. The authors responded by noting that they previously showed that blue- or yellow-light stimulation did not affect waiting times in wild-type mice (Miyazaki et al., 2014). This is an appropriate control for the experiments in Miyazaki et al. (2014), but does not address my concern here. In their previous paper, mice received a 0.8-s blue-light stimulus followed by a 50-ms yellow-light stimulus (on "blue light trials") or a 0.8-s yellow-light stimulus (on "yellow light trials"; see Fig. 2B in Miyazaki et al., 2014). In this manuscript, mice received trials in which there were two yellow-light stimuli separated by a delay during "serotonin no-activation" trials (Fig. 1b). Thus, suppose that mice thought the first of those two stimuli was actually the (first and only) yellow light on "serotonin activation" trials. They would then think the trial was over and that they did not receive reward. That is, why wait longer for the reward if you think it's not coming? I strongly recommend that the authors perform the control experiment suggested in the previous review: the same experiment in mice without the opsin in serotonin neurons (reviewer 3 and I each suggested this be done in the base strain).

2. In my previous comment (3), I suggested reporting reaction times to test whether the effects reported here could be due to behavioral inhibition, especially in light of the results of Fonseca et al. (2015). The authors responded that they previously showed that blue light stimulation did not induce behavioral inhibition in experiments 5, 12, and 13 (Miyazaki et al., 2014). I again suggest reporting this analysis, given that the task is slightly different from the previous ones in Miyazaki et al. (2014).

3. It appears the authors misunderstood my previous comment (5). I was not suggesting that the authors "threw out" data! Rather, I was asking whether the mice could have adopted a strategy in which they "threw out" low-probability sessions and performed better on high-probability sessions.

Reviewer #3 (Remarks to the Author):

With respect to the original comments

1. The addition of the model formalises the speculations in the discussion but does not in itself add any specific experimental evidence in their support. To make the model fit required "several arbitrary assumptions ... and the choice of some free parameters". The paper would be more impactful if these assumptions were able to, and were, tested experimentally.

2. This is well addressed.

3. The original problems with the statistics was the lack of correction for multiple comparisons, and the need to include all comparisons in one analysis. The authors now use ANOVA to solve these issues. However, in doing so, they have switched from using non-parametric tests (which seems appropriate) to a parametric analysis. Non-parametric 2-way ANOVA is not trivial, but perhaps checking for normality, and if necessary applying a transform would be a good idea, & for the 1-way analysis, Kruskal-Wallis may be more appropriate.

4. The use of trials rather than within-animal averages remains a problem which inflates the apparent significance of the results. The use of multiple trials of the same type is to achieve the best measure of central tendency for the data. The argument that the trials were internally consistent simply means that the results should be possible to achieve with a single trial of each type.

5. This is well addressed.

6. The fact that motivation does not appear to have changed is good to know, but does not really address the issue of whether the reward provided by a homeostatic necessity is the same as a purely hedonic reward. This is however a matter that it would be interesting to have seen some discussion of, not a critical issue with the manuscript.

7. The use of the word "patience" was a minor issue, so no point in persisting to argue over what might be seen as a pedantic point. It remains the case that by choosing to retain it in Introduction and Discussion "for motivation and interpretation of the experiment" the authors are technically implying that they are measuring an inner emotional state of annoyance in the animals, which many would dispute.

8 & 9. Addressed.

Responses to reviewers' comments

We are grateful to the reviewers for their critical comments and useful suggestions, which have helped us improve our paper. As indicated in the responses that follow, we have taken all these comments and suggestions into account in the revised version of our manuscript.

Reviewer #1

Comment #1

The manuscript is improved, but the authors did not address my main concern in the first review.

In my previous comment (1), I asked whether mice could have interpreted the first yellow light on "serotonin no-activation" trials as a trial-end cue, because yellow light was also used at the end of all trials. The authors responded by noting that they previously showed that blue- or yellow-light stimulation did not affect waiting times in wild-type mice (Miyazaki et al., 2014). This is an appropriate control for the experiments in Miyazaki et al. (2014), but does not address my concern here. In their previous paper, mice received a 0.8-s blue-light stimulus followed by a 50-ms yellow-light stimulus (on "blue light trials") or a 0.8-s yellow-light stimulus (on "yellow light trials"; see Fig. 2B in Miyazaki et al., 2014). In this manuscript, mice received trials in which there were two yellow-light stimuli separated by a delay during "serotonin no-activation" trials (Fig. 1b). Thus, suppose that mice thought the first of those two stimuli was actually the (first and only) yellow light on "serotonin activation" trials. They would then think the trial was over and that they did not receive reward. That is, why wait longer for the reward if you think it's not coming? I strongly recommend that the authors perform the control experiment suggested in the previous review: the same experiment in mice without the opsin in serotonin neurons (reviewer 3 and I each suggested this be done in the base strain).

We appreciate your suggestion for the control experiment. To examine whether 0.8 s yellow light at the onset of reward site nose poke worked as a trial-end-cue, we have conducted control experiments using wild-type mice (n = 5). Every mouse performed D3, D6 and D2-6-10 tests. We confirmed, that waiting time in the blue light trials was not significantly different from that in the yellow light trials in D3 test (for trials, $7.35 \pm$

0.11 s, $n = 239$ trials vs. 7.34 ± 0.12 s, $n = 240$ trials, $U = 28607.5$, $P = 0.96$, Mann-Whitney U test; for individual mice, $t(4) = 0.33$, $P = 0.76$, $n = 5$ mice, paired t -test). (Supplementary Fig. 3b). We also confirmed that the waiting time in the blue light trials was not significantly different from that in the yellow light trials in both D6 and D2-6-10 tests (D6 test for trials, 11.66 ± 0.15 s, $n = 240$ trials vs. 11.70 ± 0.15 s, $n = 239$ trials, $U = 28344.5$, $P = 0.83$, Mann-Whitney U test; D6 test for individual mice, $t(4) = 0.90$, $P = 0.42$, $n = 5$ mice, paired t -test; D2-6-10 test for trials, 14.57 ± 0.19 s, $n = 220$ trials vs. 14.62 ± 0.21 s, $n = 220$ trials, $U = 23938.5$, $P = 0.85$, Mann-Whitney U test; D2-6-10 test for individual mice, $t(4) = 0.39$, $P = 0.72$, $n = 5$ mice, paired t -test) (Supplementary Fig. 7d,e). These results reject the possibility that 0.8 s yellow light stimulation at the onset of reward site nose poke was interpreted as a trial-end-cue.

Comment #2

2. In my previous comment (3), I suggested reporting reaction times to test whether the effects reported here could be due to behavioral inhibition, especially in light of the results of Fonseca et al. (2015). The authors responded that they previously showed that blue light stimulation did not induce behavioral inhibition in experiments 5, 12, and 13 (Miyazaki et al., 2014). I again suggest reporting this analysis, given that the task is slightly different from the previous ones in Miyazaki et al. (2014).

To examine whether serotonin effect on promoting waiting was due to behavioral inhibition in the present light stimulation condition, we analyzed the reaction times between food reward presentation and exit from reward site. We confirmed that there was no significant difference of reaction times between serotonin activation and no-activation trials (for trials, serotonin activation, 1.61 ± 0.03 s, $n = 784$ trials vs. serotonin no-activation, 1.64 ± 0.03 s, $n = 780$ trials, $U = 321674.5$, $P = 0.45$, Mann-Whitney U test; for individual mice, $t(5) = 0.57$, $P = 0.60$, $n = 6$ mice, paired t -test). This result suggests that in our present light stimulation condition promoting waiting time by serotonin activation would not be due to behavioral inhibition.

Comment #3

3. It appears the authors misunderstood my previous comment (5). I was not suggesting that the authors "threw out" data! Rather, I was asking whether the mice could have adopted a strategy in which they "threw out" low-probability sessions and performed better on high-probability sessions.

We are very sorry for our two mistakes in previous response. First, we described "Even in the 25% one-pellet test, for each tested mouse we did not "throw out" low-reward

sessions”. The correct description was “Even in the 25% one-pellet test, for each tested mouse they did not “throw out” low-reward sessions”. Second, we described “There was no significant main effect of the number of wait errors during four consecutive recording sessions (four level; day 1, day 2, day 3, and day 4, $F(3,3312) = 1.14$, $P = 0.30$, two-way ANOVA)”. The correct description was “The time between tone site nose poke and reward site nose poke did not significantly change across four consecutive recording sessions (four level; day 1, day 2, day 3, and day 4, $F(3,3312) = 1.14$, $P = 0.30$, two-way ANOVA)”.

In addition to previous analysis, to examine whether reward probability influence mice’s strategy to perform tasks, we have calculated a food success ratio (the number of obtained foods/the number of trials for food) in both the serotonin activation and serotonin no-activation trials of 75% one-pellet and 25% one-pellet tests. For analysis across the number of tests, two-way ANOVA using serotonin activation conditions (two levels; serotonin activation and no-activations) as within-subject factors and task conditions (two levels; 75% one-pellet and 25% one-pellet tests) as between-subject factors were used. For analysis of within animal averages, two-way ANOVA using serotonin activation conditions (two levels; serotonin activation and no-activations) and task conditions (two levels; 75% one-pellet and 25% one-pellet tests) as within-subject factors were used. There was no significant main effect of the serotonin activation condition (for tests, $n = 57$ tests for 75% one-pellet, $n = 38$ tests for 25% one-pellet, $F(1,93) = 0.078$, $P = 0.78$; for individual mice, $n = 6$ mice, $F(1,5) = 0.49$, $P = 0.52$) and task condition (for tests, $F(1,93) = 0.73$, $P = 0.40$, for individual mice, $F(1,5) = 0.72$, $P = 0.44$) and no significant interaction between serotonin activation condition and task condition (serotonin activation \times task, for tests, $F(1,93) = 0.0025$, $P = 0.96$; for individual mice, $F(1,5) = 0.0055$, $P = 0.94$). This result also suggests that mice did not change their strategy during tests with high-probability and low-probability.

Reviewer #3

Comment #1

With respect to the original comments

The addition of the model formalises the speculations in the discussion but does not in itself add any specific experimental evidence in their support. To make the model fit required "several arbitrary assumptions ... and the choice of some free parameters". The paper would be more impactful if these assumptions were able to, and were, tested experimentally.

In this paper, we proposed a theoretical framework to explain our behavioral experimental results of the stronger effects of serotonin with higher probability of reward delivery and higher uncertainty of timing. We formulated a Bayesian decision model of waiting which assumes that serotonin activation increases the prior probability or subjective confidence of reward delivery. This theoretical framework was formulated after the experiment in order to coherently explain the intriguing results that serotonin stimulation is more effective for higher probability of reward delivery and higher uncertainty of reward timing. The assumption that the mice had a unimodal model of reward delivery timing even when the actual reward timings are discrete (e.g., 2, 6, 10 sec) is consistent with the unimodal distribution of the time of abandoning. It is desired to further test the validity of the model assumptions and the predictive power of the model, experimentally, but such experiments are kept as subjects of our future study.

Comment #3

The original problems with the statistics was the lack of correction for multiple comparisons, and the need to include all comparisons in one analysis. The authors now use ANOVA to solve these issues. However, in doing so, they have switched from using non-parametric tests (which seems appropriate) to a parametric analysis. Non-parametric 2-way ANOVA is not trivial, but perhaps checking for normality, and if necessary applying a transform would be a good idea, & for the 1-way analysis, Kruskal-Wallis may be more appropriate.

According to your advice, we have checked a homogeneity of variance of the waiting time ratio data in Experiment 1 and 2. Since data did not satisfied homogeneity of variance in both experiment, we have reanalyzed the waiting time ratio data in Experiment 1 and 2 using Scheirer-Ray-Hare test and Kruskal-wallis test, respectively. For multiple comparisons, Bonferroni correction was used. In Bonferroni correction for multiple comparisons, *P*-values of pairwise Mann-Whitney tests were multiplied by *m*,

where m was number of pairwise Mann-Whitney tests. Statistically significant differences were achieved when $P\text{-value} \times m < 0.05$. m was 15 and 10 in Scheirer-Ray-Hare test and Kruskal-wallis test, respectively.

Comment #4

4. The use of trials rather than within-animal averages remains a problem which inflates the apparent significance of the results. The use of multiple trials of the same type is to achieve the best measure of central tendency for the data. The argument that the trials were internally consistent simply means that the results should be possible to achieve with a single trial of each type.

We appreciate your kind advice. To compare the waiting time in serotonin activation and in serotonin no-activation by within animal averages, we have used paired t -test. These results were reported by Supplementary Figure 3 in Experiment 1 and by Supplementary Figure 7 in Experiment 2.

Comment #6

6. The fact that motivation does not appear to have changed is good to know, but does not really address the issue of whether the reward provided by a homeostatic necessity is the same as a purely hedonic reward. This is however a matter that it would be interesting to have seen some discussion of, not a critical issue with the manuscript.

We appreciate your comment. So far, we have not considered that whether provided food works as homeostatic necessity or purely hedonic reward. We keep this discussion in mind and we would like to continue the future experiment.

Comment #7

7. The use of the word "patience" was a minor issue, so no point in persisting to argue over what might be seen as a pedantic point. It remains the case that by choosing to retain it in Introduction and Discussion "for motivation and interpretation of the experiment" the authors are technically implying that they are measuring an inner emotional state of annoyance in the animals, which many would dispute.

We also appreciate your thoughtful comment. In our paper, we used the word "patience" as "patience to wait for future rewards" to explain animal's behavior. Our proposed Bayesian decision model of waiting explains "patience to wait for future rewards" may be affected by "confidence in a positive outcome" of "optimistic prediction for future reward" induced by serotonin neural activation. But we admit that there can be other interpretations.

Comment #2, #5, #8, and #9

This is well addressed.

We appreciate your comments.

Reviewers' comments:

Reviewer #1 (Remarks to the Author):

The authors have adequately addressed my comments.

Reviewer #3 (Remarks to the Author):

The authors have added a group analysis. However, they appear to have simply layered this on top of all the others. It remains unclear what the justification for showing averages of all trials (as in the original version), **and** the data for individual animals as well as the appropriate analysis. What do we learn from seeing the individual animal data? Variation is captured by the error bars.

So for instance, the first results reads

"71 In the experiment, during which 75% of the nose pokes for three seconds were
72 rewarded with one food pellet (Supplementary Fig. 1a), waiting time in the 25% of trials
73 with no reward (i.e., omission) was significantly longer with serotonin neuron activation
74 (7.92 ± 0.09 s, mean \pm s.e.m.) than without activation (6.99 ± 0.08 s; $U = 24097.5$, $P =$
75 2.45×10^{-16}) (Fig. 2a and 3). The effect was seen in each of the six mice tested ($P <$
76 0.022) (Supplementary Fig. 2), and for individual mice average waiting time with
77 serotonin neuron activation was significantly longer than without activation ($t(5) =$
78 24.05 , $P = 2.32 \times 10^{-6}$, $n = 6$ mice, paired t test) (Supplementary Fig. 3a)."

Most of this appears redundant. The section could more appropriately read.

"71 In the experiment, during which 75% of the nose pokes for three seconds were
72 rewarded with one food pellet (Supplementary Fig. 1a), waiting time in the 25% of trials
73 with no reward (i.e., omission) was significantly longer with serotonin neuron activation
74 ($t(5) = 24.05$, $P = 2.32 \times 10^{-6}$, $n = 6$ mice, paired t test) (Supplementary Fig. 3a)."

What is currently the Supplementary Figure should be the Figure in the body text and the others removed. Unless there is some critical interpretations that depends on seeing the average calculated by pooling all trials, and seeing the results for individual rats, the same approach should be applied throughout the paper.

For the other analyses, the authors have applied corrections for multiple comparisons in response to previous comments. It remains unclear that a series of paired t-test of these group data is appropriate.

The control group data added in response to a reviewer comment should be analysed together with the matching test group in a two-way comparison (i.e. an interaction plot combining current supplementary. Fig 3 a and b.

The same data are re-analysed using ratios to reduce the dimensions, thus making it easier to see the effect of the probability and reward size. This illustrates that in experiments of this type, it would be much more preferable to put the data into an appropriate mixed model analysis that would allow all the factors to be examined together, rather in the piecemeal way does here. Perhaps the approach used reflects the fact that experiments appear to have been added part way through the study so that only some animals got some of the tests.

Responses to reviewers' comments

We are grateful to the reviewers for their critical comments and useful suggestions, which have helped us improve our paper. As indicated in the responses that follow, we have taken all these comments and suggestions into account in the revised version of our manuscript.

Reviewer #1

Comment #1

The authors have adequately addressed my comments.

We appreciate your comments for improving our paper.

Reviewer #3

Comment #1

The authors have added a group analysis. However, they appear to have simply layered this on top of all the others. It remains unclear what the justification for showing averages of all trials (as in the original version), **and** the data for individual animals as well as the appropriate analysis. What do we learn from seeing the individual animal data? Variation is captured by the error bars.

So for instance, the first results reads

"71 In the experiment, during which 75% of the nose pokes for three seconds were
72 rewarded with one food pellet (Supplementary Fig. 1a), waiting time in the 25% of trials
73 with no reward (i.e., omission) was significantly longer with serotonin neuron activation
74 (7.92 ± 0.09 s, mean \pm s.e.m.) than without activation (6.99 ± 0.08 s; $U = 24097.5$, $P =$
75 2.45×10^{-16}) (Fig. 2a and 3). The effect was seen in each of the six mice tested ($P <$
76 0.022) (Supplementary Fig. 2), and for individual mice average waiting time with
77 serotonin neuron activation was significantly longer than without activation ($t(5) =$
78 24.05 , $P = 2.32 \times 10^{-6}$, $n = 6$ mice, paired t test) (Supplementary Fig. 3a)."

Most of this appears redundant. The section could more appropriately read.

"71 In the experiment, during which 75% of the nose pokes for three seconds were
72 rewarded with one food pellet (Supplementary Fig. 1a), waiting time in the 25% of trials
73 with no reward (i.e., omission) was significantly longer with serotonin neuron activation
74 ($t(5) = 24.05$, $P = 2.32 \times 10^{-6}$, $n = 6$ mice, paired t test) (Supplementary Fig. 3a)."

What is currently the Supplementary Figure should be the Figure in the body text and the others removed. Unless there is some critical interpretations that depends on seeing the average calculated by pooling all trials, and seeing the results for individual rats, the same approach should be applied throughout the paper.

We appreciate your detailed suggestions. We removed averages of all trials from the main text and showed results of group analysis of Experiment 1 and Experiment 2 in Fig.3 and Fig.6 of the revised manuscript, respectively. Because the waiting time histograms of all omission trials are valuable information to capture mice's general behaviors, we kept those of Experiment 1 and Experiment 2 in Fig.2 and Fig.5 of the revised manuscript, respectively. For reference, averages of all trials of Experiment 1 and Experiment 2 are shown in Supplementary Fig.2 and Supplementary Fig.7 of the revised manuscript,

respectively.

The reason to show the data of individual mice is to confirm whether optogenetic activation of serotonin neurons have an effect on waiting behavior to each of tested mice. We showed that in all tested mice optogenetic serotonin activation induced significant same effect on waiting for delayed reward at various test conditions. This information is not available in group analysis. In the revised manuscript, data of individual mice in Experiment 1 are shown in Supplementary Fig.3 and Supplementary Fig.5. Data of individual mice in Experiment 2 are shown in Supplementary Fig.8 and Supplementary Fig.9.

Comment #2

For the other analyses, the authors have applied corrections for multiple comparisons in response to previous comments. It remains unclear that a series of paired t-test of these group data is appropriate.

In previous revision, we have reanalyzed the waiting time ratio data in Experiment 1 and 2 using Scheirer-Ray-Hare test and Kruskal-wallis test, respectively. For multiple comparisons, Bonferroni correction was used. In Bonferroni correction for multiple comparisons, *P*-values of pairwise Mann-Whitney *U* tests, not those of paired t-tests, were used.

Comment #3

The control group data added in response to a reviewer comment should be analysed together with the matching test group in a two-way comparison (i.e. an interaction plot combining current supplementary. Fig 3a and b.

According to your suggestion, we reanalyzed control group (WT) data with ChR2 expressing group (ChR2) in a two-way ANOVA. The result of the 75% one-pellet test (equivalent the D3 test) was shown in Fig. 3a. The results of D6 and D2-6-10 tests were shown in Fig. 6a and 6c, respectively. In three tests, there was a significant main effect of light (two levels within-subject factors; yellow and blue) but no significant main effect of group (two levels between-subject factors; ChR2 and WT). Since there was a significant main effect of interaction (light \times group), we analyzed a simple main effect of light in ChR2 and WT. There was a significant simple main effect of light in ChR2 but no significant simple main effect of light in WT.

Comment #4

The same data are re-analysed using ratios to reduce the dimensions, thus making it easier

to see the effect of the probability and reward size. This illustrates that in experiments of this type, it would be much more preferable to put the data into an appropriate mixed model analysis that would allow all the factors to be examined together, rather in the piecemeal way does here. Perhaps the approach used reflects the fact that experiments appear to have been added part way through the study so that only some animals got some of the tests.

Analysis of Fig.3 and Fig.4 (Experiment 1 data) were different analysis. In Fig.3 we showed how absolute value of waiting times were enhanced by serotonin activation. However, this analysis could not show how effectively serotonin activation prolong waiting times. To quantify the effectiveness of serotonin activation on waiting time, we calculated the waiting time ratio in each test, the number of which was from 5 to 13 in one reward condition, and this calculation did not simply reduce the dimensions. Same procedures applied to Experiment 2 data (Fig. 6).

In addition to non-parametric two-way ANOVA, we have performed additional analysis based on a linear mixed model, taking into account individual differences among mice. This analysis allows all the factors including levels to be examined together as you pointed out. The results of the analysis suggest that the difference of means between 75% reward probability and the remainder of probability is significant while the other combinations are not. This supports the results of the subsequent analysis in Fig.4. As regards the imbalance of data, three mice did not experience 50 % reward probability tests, but it was in a random manner. Since the variability of waiting time ratio among mice was not significant ($P = 0.553$), the absence of the data related to these mice does not bias the results of our analysis. Similarly, we carried out a mixed-model analysis for Experiment 2 data.

REVIEWERS' COMMENTS:

Reviewer #3 (Remarks to the Author):

The authors have worked to address the statistical issues raised

Responses to reviewers' comments

We are grateful to the reviewers for their critical comments and useful suggestions, which have helped us improve our paper.

Reviewer #3

Comment #1

The authors have worked to address the statistical issues raised

We appreciate your comments for improving our paper.